# ΔFosB accumulation in hippocampal granule cells drives cFos pattern separation during spatial learning

Paul J. Lamothe-Molina [1,6] ✉, Andreas Franzelin [1,6], Lennart Beck[1], Dong Li[2], Lea Auksutat[3], Tim Fieblinger [1], Laura Laprell [1], Joachim Alhbeck[4], Christine E. Gee [1], Matthias Kneussel [5], Andreas K. Engel [4], Claus C. Hilgetag[2], Fabio Morellini[3,7] & Thomas G. Oertner [1,7] ✉

Mice display signs of fear when neurons that express cFos during fear conditioning are artificially reactivated. This finding gave rise to the notion that cFos marks neurons that encode specific memories. Here we show that cFos expression patterns in the mouse dentate gyrus (DG) change dramatically from day to day in a water maze spatial learning paradigm, regardless of training level. Optogenetic inhibition of neurons that expressed cFos on the first training day affected performance days later, suggesting that these neurons continue to be important for spatial memory recall. The mechanism preventing repeated cFos expression in DG granule cells involves accumulation of ΔFosB, a long-lived splice variant of FosB. CA1 neurons, in contrast, repeatedly expressed cFos. Thus, cFos-expressing granule cells may encode new features being added to the internal representation during the last training session. This form of timestamping is thought to be required for the formation of episodic memories.

Adaptive decision making requires an accurate memory of past events. For instance, a mouse should revisit locations where it has found food before and avoid dangerous places. The hippocampus is well-known for processing spatial information[1,2] but it also processes other types of sensory input and organizes sequential elements of experience into episodic memories[3–5]. Our understanding of how time is represented in the hippocampus is incomplete. The expression of many genes affecting synaptic plasticity undergoes pronounced circadian oscillations, changing the rules of synaptic plasticity depending on the time of day[6]. Theoretical and empirical work suggests that the dentate gyrus (DG) actively reduces the overlap between activity patterns from the entorhinal cortex, a process dubbed pattern separation[7–9]. In vivo electrophysiology and calcium imaging experiments revealed that the majority of the granule cells (GCs) in the DG are silent, and of the active cells, just a small fraction show spatial tuning (place cells)[10–12]. Spatially tuned GCs provide a stable, albeit coarse representation of the global environment across time[10,13–15]. These findings highlight an apparent design conflict: while a perfect 'episode encoder' should avoid using the same neurons on consecutive days, accurate place coding is thought to require stable place cells that signal the animal's position in a given environment. In view of this conundrum, we set out to study the impact of time and space on GCs in the dorsal DG.

[1]Institute for Synaptic Physiology, Center for Molecular Neurobiology (ZMNH), University Medical Center Hamburg-Eppendorf, Hamburg, Germany. [2]Institute of Computational Neuroscience, University Medical Center Hamburg-Eppendorf, Hamburg, Germany. [3]Research Group Behavioral Biology, Center for Molecular Neurobiology (ZMNH), University Medical Center Hamburg-Eppendorf, Hamburg, Germany. [4]Department of Neurophysiology and Pathophysiology, Center for Experimental Medicine (ZEM), University Medical Center Hamburg-Eppendorf, Hamburg, Germany. [5]Institute for Molecular Neurogenetics, Center for Molecular Neurobiology (ZMNH), University Medical Center Hamburg-Eppendorf, Hamburg, Germany. [6]These authors contributed equally: Paul J. Lamothe-Molina, Andreas Franzelin. [7]These authors jointly supervised this work: Fabio Morellini, Thomas G. Oertner. ✉e-mail: paul@lamothe.de; thomas.oertner@zmnh.uni-hamburg.de

To monitor and manipulate neuronal activity in freely behaving animals, reporter mice have been developed that use the immediate-early gene cFos[16] to drive the expression of fluorescent proteins and optogenetic actuators[17]. Fear conditioning experiments with an activity-dependent expression of optogenetic silencing tools suggest that recall of a fearful episode requires reactivation of the original encoding ensemble[18–20]. Vice versa, artificial reactivation of the encoding ensemble has been shown to reinstate a fearful state[19,21,22], suggesting that memories can be activated by specific subsets of hippocampal neurons. When freezing is used as a proxy to assess the emotional state of the animal, it is difficult to determine whether, during the optogenetic reactivation, the animal recalls the fear-inducing episode (engram) or simply feels fear. The reactivated ensemble may form an internal representation of the external world (cognitive map theory[23,24]) or trigger the pattern of cortical activity that was active in the previous experience (indexing theory[25]).

We investigated the temporal stability of DG cFos ensembles and their relation to spatial learning and the formation of cognitive maps. We used cFos-dependent tagging to assess how cFos ensemble overlap changes over training days in the Morris water maze (WM). Even in expert mice, cFos expression patterns in DG changed from day to day. In spite of these changing expression patterns, optogenetic inhibition of cFos-tagged GCs impaired navigation 5 days after tagging, suggesting that the absence of cFos expression in these GCs did not imply the absence of activity. We show in vivo and in vitro that cFos$^+$ GCs accumulate ΔFosB, a long-lived splice variant of FosB. As this splice variant inhibits cFos expression[26], it provides a potential mechanism for the daily shift of cFos ensembles that occurs even when mice are exposed to the same environment. The changing pattern of cFos expression in DG could be the basis of an episodic memory system that write-protects synapses on the most recently used subset of GCs during the following days.

## Results

### Spatial learning and cFos ensemble overlap in the DG: novice vs expert mice

Although cFos expression is considered a proxy of neuronal activity[27–29], the relationships between action potential firing, immediate-early gene expression, and synaptic long-term plasticity are not obvious[30]. We chose a spatial learning paradigm, the Morris WM, to investigate cFos expression patterns in DG during learning. TetTag mice were trained to find a hidden platform in the WM (Fig. 1a–c). We labeled cFos ensembles from two consecutive days (Fig. 1d). The window for labeling the first cFos ensemble was opened by taking the mouse off doxycycline (Dox), allowing for long-term expression of the fluorescent protein mKate2 fused to an opsin for membrane labeling (cFos-tagged, Fig. 1d). A short half-life version of the green fluorescent protein (shEGFP) works as an in-built cFos reporter in the TetTag mouse to label the second cFos$^+$ ensemble. In calibration experiments, we observed slightly faster temporal dynamics of the native cFos protein than of the shEGFP reporter, yet nearly all shEGFP-expressing neurons also expressed cFos (Supplementary Fig. 1). The highest correlation between native cFos and shEGFP was obtained 2–4 h after stimulation. We, therefore, chose 3 h after the last training trial as the time point to sacrifice mice and fix the brains. The first cFos-tagged ensemble was revealed by immunofluorescence (IF) staining against mKate2 and the second cFos-expressing ensemble was revealed by staining against the shEGFP (cFos$^+$). The ensemble sizes (proportion of GCs cFos-tagged and cFos$^+$, Supplementary Fig. 2) were used to calculate the amount of overlap (GCs that are both cFos-tagged and cFos$^+$) expected due to chance for each sample and compared to the actual overlap (number of cFos-tagged cFos$^+$ GCs).

During probe trials, mice early in training (ET, Fig. 1b, c, e, f) were less precise and accurate while searching for the target but within 5 days became experts (overtrained group, OT, Fig. 1b, c, g, h). Not only

the time spent searching in an annulus around the hidden platform increased (Fig. 1f, h) but also the distance to the platform was significantly lower in the expert mice (Fig. 1i). Our expectation was that, if cFos expression is linked to the activity of the neurons that are important for solving the WM task, there would be a high degree of overlap between the cFos$^+$ and the cFos-tagged neurons, which were labeled 1 day earlier. We also expected overlap to increase with increasing search accuracy in the probe trials. Instead, we observed that very few GCs expressed both mKate2 (cFos-tagged) and shEGFP (cFos$^+$) and that both overlap (ET 9% and OT 7% Fig. 1e, g, j) and ensemble sizes were similar in novices and expert mice (Supplementary Fig. 2). In both ET and OT mice the observed overlap was, however, significantly higher than expected due to chance (Fig. 1j, k, Supplementary Fig. 2). Surprisingly, there was no difference between the novices and experts and, assuming that cFos indicates the active GCs, less than 90% of the neurons that expressed cFos the first day were reactivated the following day. Thus, in the DG cFos overlap in GCs on consecutive days of WM training is low and independent of spatial search accuracy.

### Effect of novelty on cFos ensembles: reversal training vs novel environment

We next tested whether changing the position of the escape platform (reversal training) would decrease cFos ensemble overlap. The first cFos ensemble was labeled in expert mice on day 5 of WM training (OFF Dox Fig. 2a). The following day, these mice were trained (ON-Dox) with the platform on the opposite side of the tank (reversal training, RT). Reversal training with the new platform position had no effect on cFos overlap (9%, Fig. 2a; compare with ET and OT groups, Fig. 1e, g), although it dramatically decreased search accuracy during the day 6 probe trials (Fig. 2b). The mice now searched equally in both E and W annuli, further suggesting that cFos overlap in DG GCs is a poor indicator of performance.

Given that after reversal training, the overlap was still higher than expected (Fig. 2f), we tested whether overlap of cFos ensembles is specific to the WM. The overlap was reduced to 4% and no longer above chance when cFos-tagged ensemble on day 1 of WM training was compared with the cFos+ ensemble generated while mice explored an open field for 20 min (Fig. 2c, d, f, novel environment NE). Likewise, cFos overlap was only 2% if mice were kept in their home cages (Fig. 2e, f home cage HC) for the 24 h OFF Dox, as in the other groups, and mice were sacrificed the following day. The cFos overlap of all groups trained in the WM (the ET, OT, and RT mice) were not significantly different from each other but significantly higher than expected due to chance whereas in both NE and HC groups, cFos overlap was at chance level and significantly lower than the WM groups (Fig. 2f, Supplementary Fig. 2). Despite ensemble sizes being the same in all conditions except the home cage, the intensity of shEGFP was higher in all groups WM trained on the second day (Fig. 2g, Supplementary Fig. 2). Thus, DG cFos expression and overlap are strongly driven in the WM regardless of training level (ET vs OT) or difficulty of the task (OT vs RT), consistent with the concept that the hippocampus is highly engaged in spatial navigation tasks and that very different environments activate non-overlapping ensembles.

### cFos$^+$ neurons in DG participate in spatial memory recall days later

Lesion[31,32] and optogenetic manipulation[33–36] experiments indicate that the DG plays a role in spatial memory acquisition and recall. However, those experiments did not address if particular memories are encoded by a specific subset or ensemble of neurons in the DG (engram cells). We speculated that cFos$^+$ cells in the DG may encode relevant information needed to solve the WM task. To test their importance during platform search, we employed a bidirectional optogenetic tool that can be used to inhibit or excite neurons with 473 and 594 nm light,

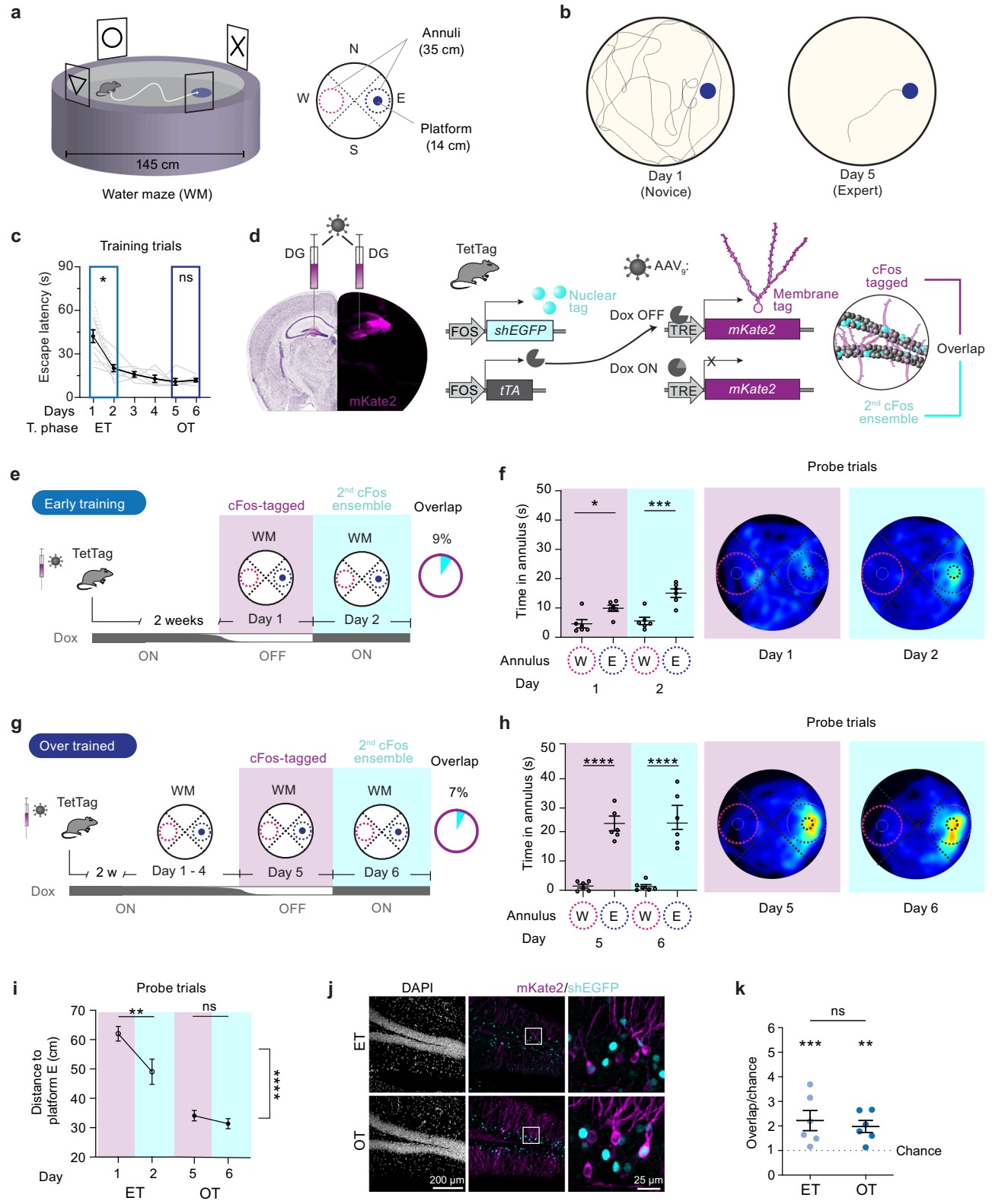

respectively (BiPOLES[37]). We expressed BiPOLES under the control of the non-leaky TRE3G promoter (Supplementary Fig. 3) and confirmed that 473 nm light prevented action potentials in BIPOLES-expressing GCs while 594 nm light pulses repeated at 20 Hz reliably elicited single action potentials (Supplementary Fig. 4). For in vivo experiments, mice were bilaterally injected in DG with AAV_{PHP.eB}-TRE3G-*BiPOLES-mKate2* and implanted with a custom-made tapered fiber implant[38] (Fig. 3a). To perform optogenetic experiments in the WM, we had to ensure that

the weight of the optical fibers was compensated when mice were tethered to them via their implants. We achieved this by attaching a helium balloon to the optical fibers (Fig. 3a). Implanted mice were trained without being tethered and successfully learned the WM task (Fig. 3b). Tethering to the weight-balanced optical fibers did not affect swimming speed (Fig. 3c), nor did 473 or 594 nm light (Fig. 3d).

To investigate the importance of day 1 cFos-tagged neurons on memory recall, we silenced BiPOLES-expressing neurons during

**Fig. 1 | Behavioral performance in a spatial memory task is not reflected by cFos ensemble overlap in the DG. a** Water maze (WM) with the platform in east quadrant (E). Virtual 35 cm diameter annuli for spatial accuracy analysis (E vs W). **b** Exemplary mouse swim paths during training trials. **c** WM learning (escape latency) over 6 training days (black line: average of all mice. days 1–2, $n = 12$; days 3–6, $n = 6$). Note the rapid learning during early training (ET, $n = 6$ mice, $*p = 0.02$) and lack of improvement when overtrained (OT, $n = 6$ mice, $n = 0.98$). Mixed effects model with Šídák's multiple comparisons test. **d** Tagging method. In TetTag mice injected with AAV-TRE-*mKate2*, cFos+ granule cells (GC) express mKate2 in the absence of Doxycycline (Dox), resulting in permanent fluorescence (cFos-tagged, magenta). Left hemisphere: Nissl from the Allen Reference Atlas[70]. Nuclear fluorescence identifies recent cFos expression (2nd ensemble, shEGFP, cyan). Overlap is the fraction of mKate2+ neurons expressing shEGFP. **e** Experimental timeline for ET mice and cFos overlap. **f** Time spent in annuli in probe trials on day 1 and 2 (ET

mice). Heatmaps show average swim paths during probe trials ($n = 6$ mice, $*p = 0.03$, $***p = 0.001$). **g** Experimental timeline for OT mice and cFos overlap. **h** Time spent in annuli in probe trials on day 5 and 6. Heatmaps show average swim paths during probe trials ($n = 6$ mice, $****p < 0.0001$). **i** Spatial search accuracy (distance to platform E) was significantly higher in the OT than in the ET group ($****p < 0.0001$). Only ET mice improved their spatial accuracy between tagging days (ET group: $**p = 0.004$; OT group: ns $p = 0.65$). **f, h, i** Symbols represent individual mice, lines represent mean ±SEM. Matched two-way-ANOVA with Šídák's multiple comparisons test. **j** Immunofluorescence images for cFos overlap analysis. **k** cFos overlap significantly higher than chance: ET: $***p = 0.0008$, OT: $**p = 0.003$, see Supplementary Fig. 2) but similar in both groups ($p = 0.99$, ordinary one-way-ANOVA with Šídák's multiple comparisons test). Data are presented as mean ±SEM. Complete statistical information in Supplementary Table 1. Source data are provided as a Source Data file.

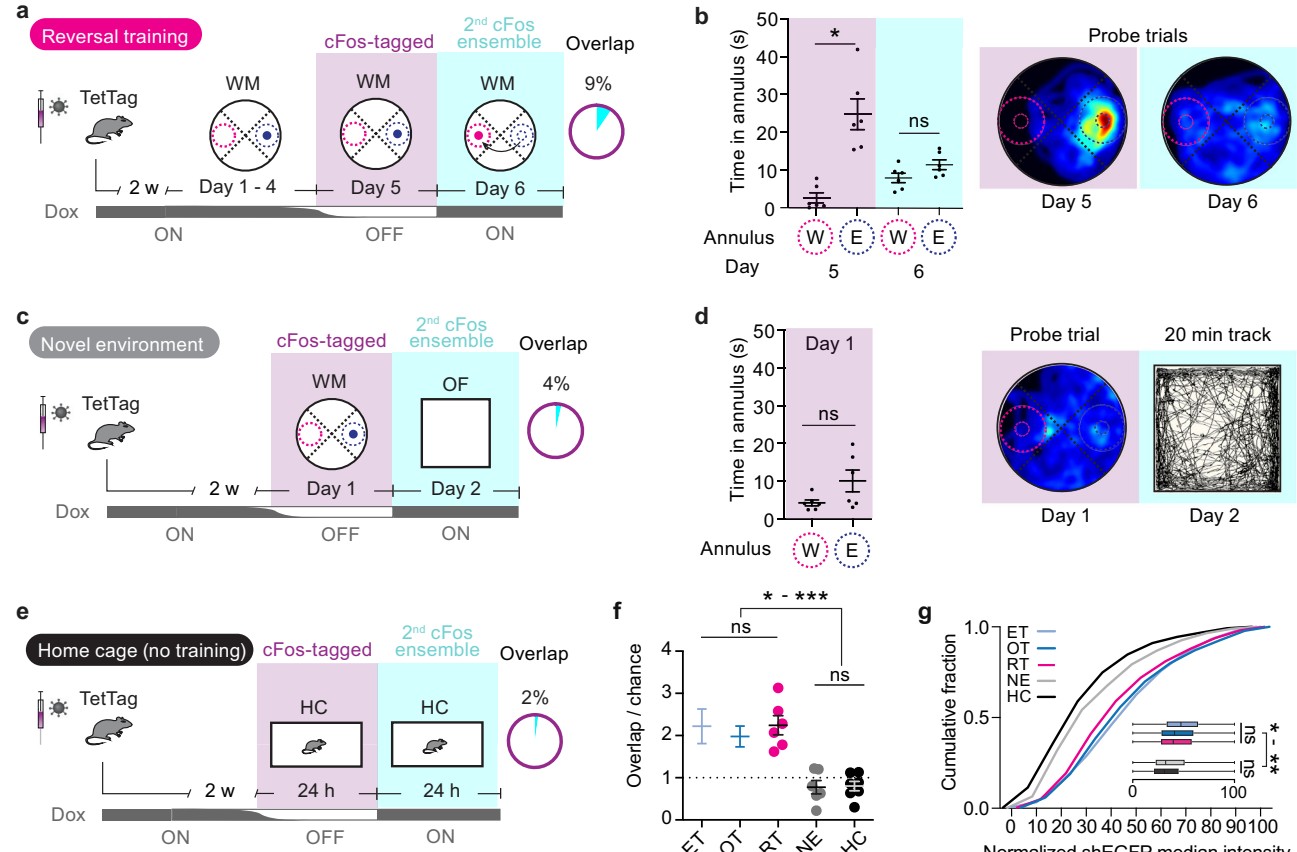

**Fig. 2 | Effects of task and environment on cFos ensemble overlap.**
**a** Experimental timeline for reversal training (RT) group ($n = 6$ mice) and cFos overlap. Note that the platform position was changed to the opposite quadrant (W) on day 6. **b** Time spent in annuli in probe trials on day 5 and 6 of RT mice. Heatmap showing average swim paths during probe trials. Note that despite the strong preference for E annulus on day 5 ($n = 6$ mice, $*p = 0.01$), after only 1 reversal training session, mice searched equally around the new and the old platform position during a probe trial on day 6 (ns, $p = 0.72$). Matched two-way-ANOVA with Šídák's multiple comparisons test. **c** Experimental timeline for novel environment (NE) group and cFos overlap. Note that mice visit an unfamiliar open field (OF) arena on day 2. **d** Time NE mice spent in annuli during day 1 probe trial shows no significant preference for E annulus ($n = 6$ mice, $p = 0.16$, two-sided, paired $t$ test). Heatmap shows average swim paths during probe trial on day 1. Exemplary track of

one mouse during OF exploration on day 2. **b, d** Symbols represent individual mice, lines represent mean ±SEM. **e** Experimental timeline for home-caged (HC) control mice and cFos overlap. **f** cFos ensemble overlap/chance in the different experimental groups (ET early training from Fig. 1i; OT, overtrained from Fig. 1i; RT reversal training, NE novel environment, HC home cage). cFos overlap in the RT group is higher than expected by chance ($n = 6$ mice, $***p = 0.0006$, matched two-way-ANOVA with Šídák's multiple comparisons test, see Supplementary Fig. 2). All WM-trained groups showed higher cFos overlap than the NE and HC groups ($p$ values in supplementary table 1), but no difference between each other. Symbols represent individual mice, lines represent mean ±SEM. **g** cFos expression (normalized shEGFP median intensity) was significantly higher in mice that visited the WM than mice that visited an OF arena (NE group) or control mice (HC group) ($p$ values in Supplementary table 1). Source data are provided as a Source Data file.

probe trials on days 2, 3, and 5 (Fig. 3f). Mice were placed on an elevated Atlantis platform that submerged after 30 s, forcing the mice to swim[39]. After 60 s, a second Atlantis platform (in quadrant E) was elevated, and mice were directed towards it if they did not find it themselves. We assessed memory performance in the first

half of the probe trials (30 s). Optogenetic inhibition did not affect time spent in the target quadrant on days 2 and 3 when mice were novices and their spatial accuracy was still poor (Fig. 3g). On day 5, when mice were experts and spatial accuracy was high, optogenetic inhibition significantly decreased time spent in the target

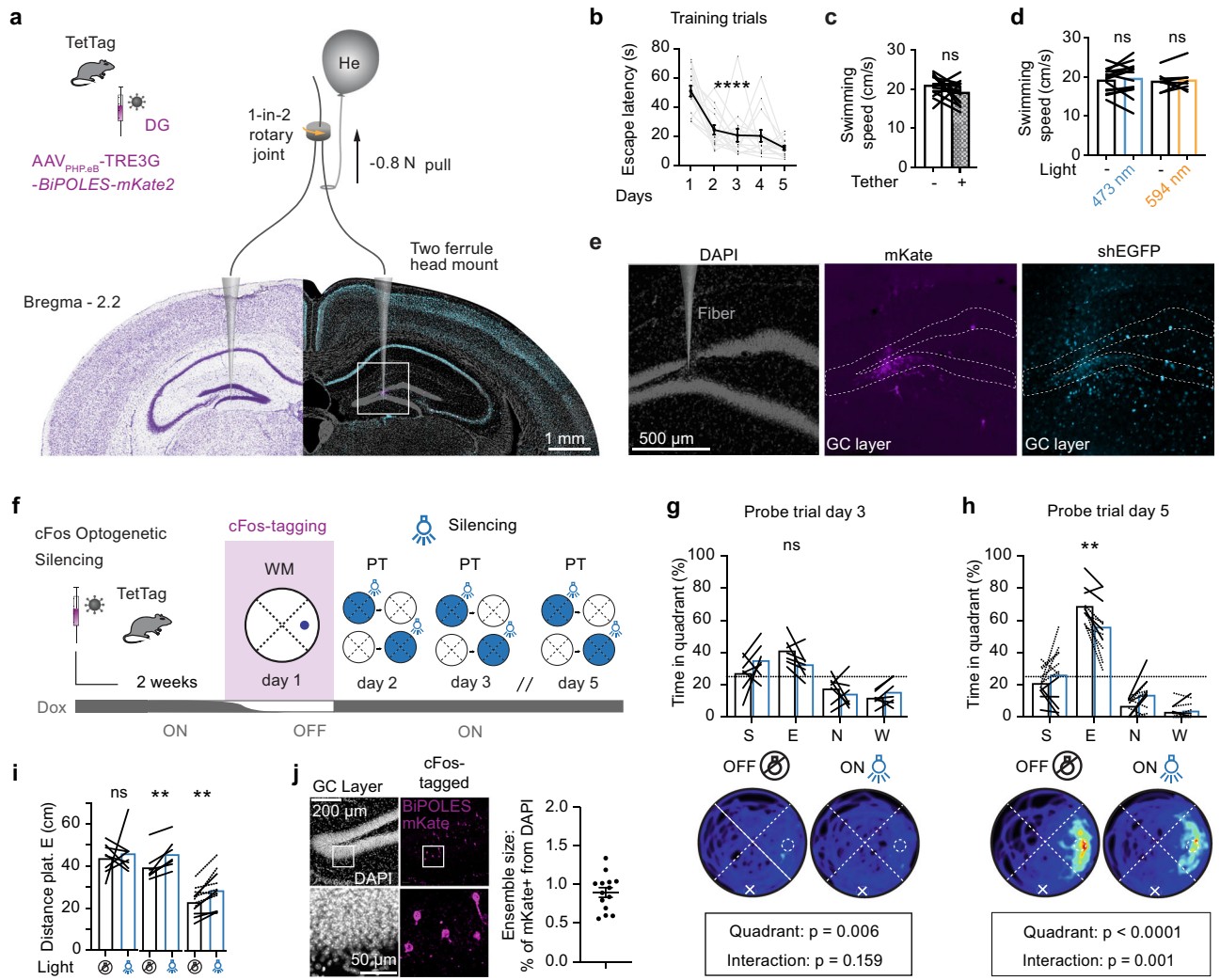

**Fig. 3 | Optogenetic silencing of cFos-tagged granule cells impairs spatial memory recall. a** TetTag mice were injected with AAV for cFos-dependent expression of BiPOLES and fluorescent label (mKate2, magenta). Mice were bilaterally tethered with ultra-thin light fibers targeting the granule cell (GC) layer of the dentate gyrus[70]. **b** Fibers and implant did not affect learning (gray lines: $n = 15$ mice; black: mean ±SEM; ****$p < 0.0001$, ordinary one-way-ANOVA for repeated measures). **c** Swimming speed of tethered mice with balloons was not different from the same animals in untethered trials ($n = 15$ mice, $p = 0.12$, two-sided paired $t$ test). **d** Optogenetic inhibition (473 nm) or excitation (596 nm) did not affect swimming speed (473 nm light, $n = 15$ mice, $p = 0.53$, two-sided paired $t$ test) (594 nm light, $n = 8$ mice, $p = 0.74$, paired $t$ test). **e** Tapered fiber caused minimal damage in DG. **f** On days 2, 3, and 5 of water maze training, the cFos-tagged ensemble from day 1 was inhibited (blue shading) in one of two tethered probe trials. **g** On day 3, probe trial optogenetic inhibition had no significant effect on the time spent in the target quadrant ($p = 0.35$, matched two-way-ANOVA with Šídák's multiple comparisons test). Heatmaps show average swim paths during probe trials. White 'x' indicates starting Atlantis platform location, dashed circle indicates the location of target Atlantis platform. **h** On day 5, probe trial optogenetic inhibition significantly decreased the time spent in the target quadrant (Light x Quadrant interaction ***$p = 0.001$; light effect **$p = 0.0025$). Similar results were obtained in two batches of mice (solid/dotted lines). **i** Optogenetic inhibition increased the average distance from the target platform location on days 3 and 5, but not on day 2 (two-sided, paired $t$ test: day 3 **$p = 0.004$, Day 5 **$p = 0.002$). **j** After the behavioral experiments, the expression of BiPOLES was verified in 14 of 15 animals ($n = 14$ mice, mean ±SEM). **c**, **d**, **g**–**i** Lines show individual mice, bars show mean. Complete statistical information in Supplementary Table 1. Source data are provided as a Source Data file.

quadrant (Fig. 3h, Supplementary Video 1). The mean distance to the target platform confirmed the results of the quadrant analysis, detecting significant effects of optogenetic inhibition on days 3 and 5 (Fig. 3i). We confirmed that GCs still expressed BiPOLES even 9 days after tagging (Fig. 3j). These results suggest that activity in the small ensemble of cFos-tagged neurons from the first training day (~2% of GCs) is important for successful memory recall on subsequent days. BiPOLES can also be used to increase the activity of tagged neurons (Supplementary Fig. 4). However, artificial firing (20 Hz optogenetic stimulation) during day 6 probe trials had no significant effect on WM performance in expert mice (Supplementary Fig. 5).

To address whether the activity of cFos-tagged cells is of particular importance for WM navigation or if inhibition of a random group of GCs would suffice to decrease performance, we used a chemogenetic inhibition strategy that is not dependent on cFos (Supplementary Fig. 6). The proportion of hM4Di-mCherry expressing neurons in the DG determined whether chemogenetic silencing disrupted memory recall ($R^2 = 0.67$, $p = 0.003$, Supplementary Fig. 6). Silencing of ~30% of GCs was required to decrease WM performance, a much higher fraction compared to cFos-dependent optogenetic silencing, suggesting that GCs expressing cFos on training day 1 remain important for successful spatial navigation in the WM many days later. If cFos-tagged cells from day 1 are indeed highly active on consecutive WM days, this

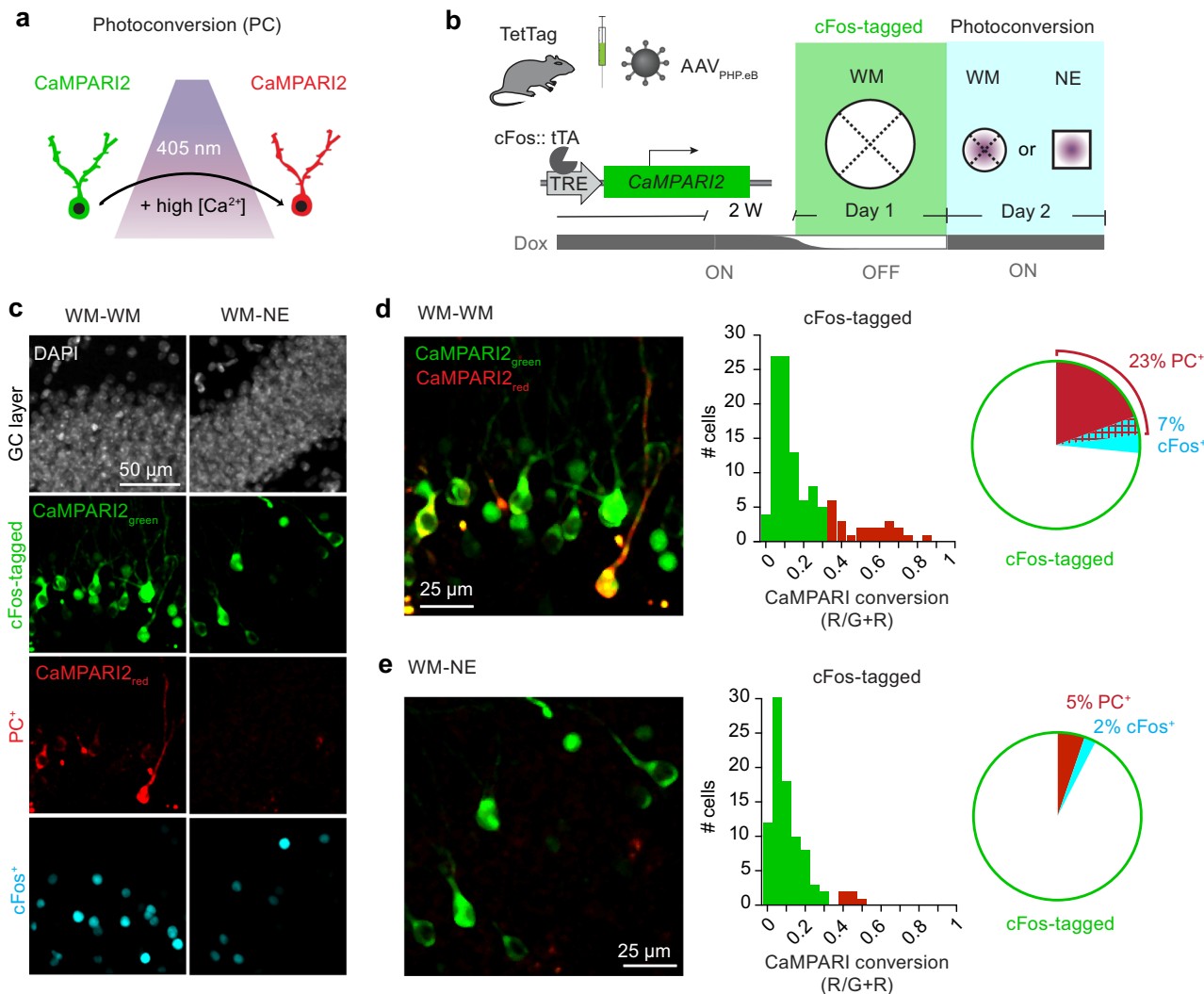

**Fig. 4 | cFos-tagged dentate granule cells are reactivated during water maze training. a** The calcium integrator CaMPARI2 is photoconverted by 405 nm light (violet shading) only in neurons with high intracellular $Ca^{2+}$. **b** Experimental timeline: TetTag mice were injected with $AAV_{PHP.eB}$-TRE-*CaMPARI2* and trained in the WM OFF Dox (day 1, cFos tagging, green shading). On day 2, photoconversion light was applied to the DG during WM training or in a novel environment (NE, open field). **c** Confocal images of DG. Gray is DAPI staining, green is cFos-dependent CaMPARI2 expression, red is immunostaining against photoconverted CaMPARI2 (PC+), cyan is shEGFP immunostaining (cFos+) performed 90 min after the last trial.

Note: CaMPARI2 is expressed throughout the cytosol, green nuclear signal is the native shEGFP cFos reporter protein from the TetTag mice. **d, e** Analysis of CaMPARI2 conversion. Left: Merged images of CaMPARI2 (green) and photoconverted CaMPARI2 (red). Histogram: red bars indicate photoconverted neurons (ratio >0.3). Pie charts indicate % of cFos-tagged (CaMPARI2 expressing) neurons that were photoconverted (red, PC+), cFos+ (cyan), or both (red hatched). **d** Day 1 and day 2 in water maze (WM-WM, *n* = 113 cells, 2 mice). **e** Day 1 WM day 2 novel environment (WM-NE *n* = 100 cells, 2 mice). Source data are provided as a Source Data file.

activity should generate intracellular calcium transients. To generate a lasting record of calcium concentrations during behavior, we tagged cFos+ GCs with calcium-dependent photoconvertible CaMPARI2[40] on training day 1 and applied photoconversion light via implanted optical fibers either during all WM training trials on day 2 (WM–WM) or during the same amount of time in a novel environment (WM–NE, Fig. 4a, b). In order to estimate the volume of tissue that can be photoconverted by 405 nm light, we tagged cFos+ GCs with CaMPARI2 together with an opsin that is activated by PC light, resulting in strong photoconversion of cells up to 300 μm from the tapered fiber tip (Supplementary Fig. 7). Based on these tests, we restricted our analysis to GCs within 200 μm of the fiber tip to assess calcium levels during behavior. During WM training on day 2, photoconversion occurred in 23% of the cFos-tagged GCs, indicating a high degree of functional reactivation (Fig. 4d). This reactivation was context-specific, as only 5% cFos-tagged GCs were photoconverted during exploration of a novel environment (Fig. 4e)[41]. We also analyzed cFos expression on day 2 in the cFos-tagged cells. As

expected from our previous experiments (Figs. 1 and 2), cFos overlap was higher in the WM-WM group (7%, Fig. 4d) than in the WM-NE group (2%, Fig. 4e). Interestingly, even in the WM-WM group, 85% of photoconverted (and thus reactivated) GCs did not express cFos again, confirming our suspicion that cFos expression is not a good proxy for GC activity.

## Temporal dynamics of cFos ensembles

An engram cell, by definition, is a cell that is active during both the encoding event and during memory retrieval[17]. Many studies used cFos expression as a surrogate marker of neuronal activity, but our WM experiments suggest that the underlying assumption may be wrong. In our experiments, cFos overlap was about 9% when mice revisited the WM arena on consecutive days ($\Delta t$ = 1 days). Would these GCs continue to activate cFos on the following days of WM training? To evaluate longer time intervals, we trained the mice exactly as the RT group (highest overlap) but tagged GCs on day 1 and sacrificed the mice on

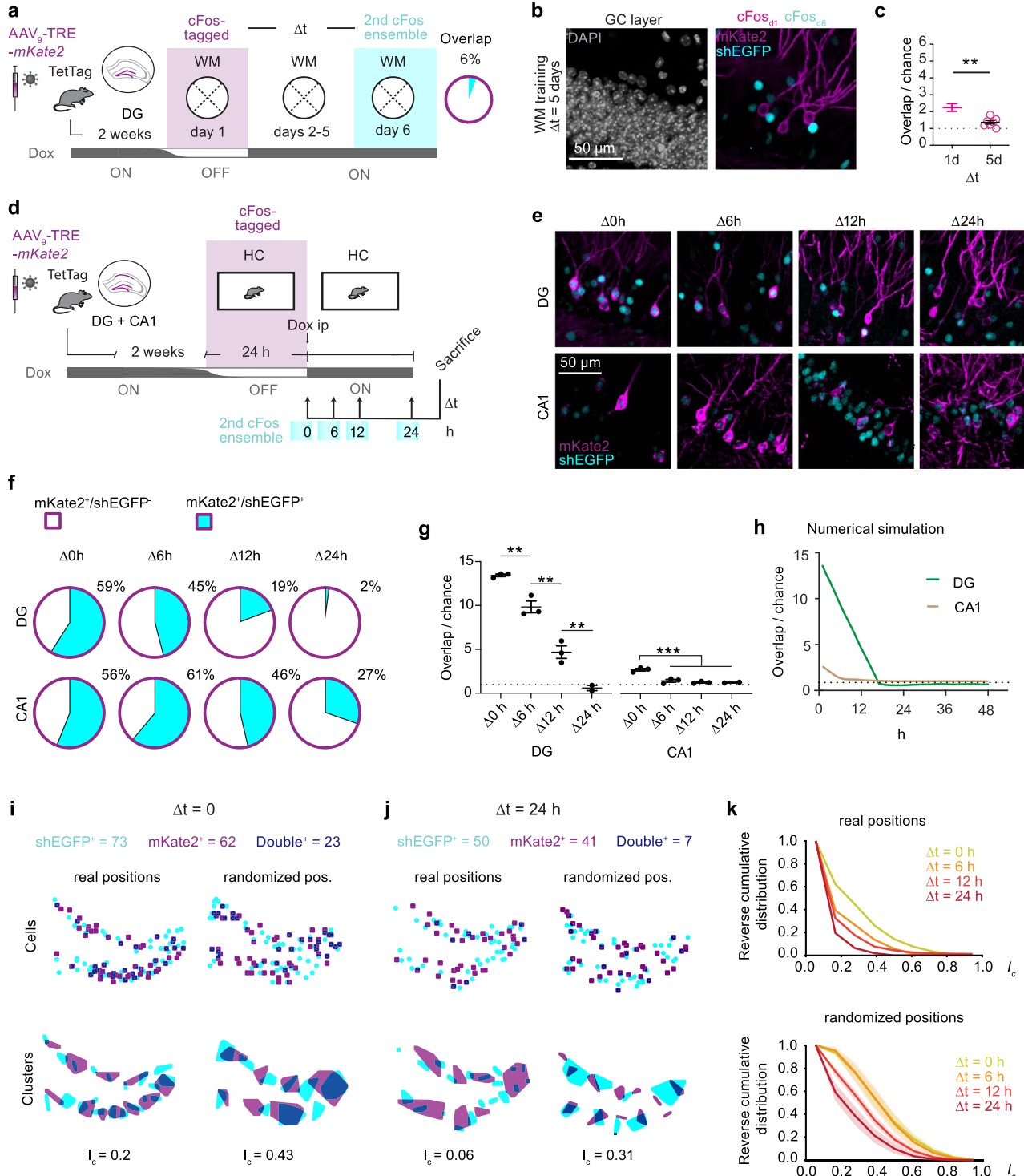

**Fig. 5 | Temporal dynamics of cFos overlap in dentate gyrus. a** Experimental timeline and cFos overlap. TetTag mice were injected with $AAV_9$-TRE-*mKate2* in dentate gyrus (DG) to tag cFos$^+$ neurons on training day 1 (magenta shading, cFos$_{d1}$). They were trained for 6 days and sacrificed for shEGFP immunostaining (cyan shading, cFos $_{d6}$). **b** Immunofluorescence example images from the granule cell (GC) layer showing low overlap between cFos$_{d1}$ (magenta) and cFos$_{d6}$ (cyan). **c** After 5 days of WM training, cFos overlap was not significantly different from chance ($n = 7$ mice, two-way-ANOVA, $p = 0.11$, see Supplementary Figure 2) and significantly lower (**$p = 0.003$, two-sided, unpaired $t$ test) than after 1 day of WM training (RT group, data from Fig. 2d). **d** Home cage experiment. TetTag mice were injected AAV injected with $AAV_9$-TRE-*mKate2* in DG and CA1 regions. Dox was removed to tag cFos neurons while mice were in their home cage (HC). Mice were sacrificed at different time points after Dox injection. **e** IF staining of cFos-tagged and shEGFP (cFos$^+$) neurons in DG and

CA1. **f** Fractions of cFos-tagged cells that remained cFos-positive for different time intervals Δt (mean values, three mice per group). **g** Detected cFos overlap normalized to the expected overlap for random expression (dotted line: chance level) in DG and CA1 ($n = 3$ mice per group, mean ± SEM, ordinary one-way-ANOVA with Šídák's multiple comparisons test). See Supplementary Table 1 for full statistical information. **h** Numerical simulation of cFos expression with negative feedback (green curve) reproduces linear drop in cFos pattern similarity found in DG. CA1 simulation without negative feedback (brown curve) does not drop below 1 (chance level). **i** Clusters of cFos$^+$ cells have less overlap ($I_c$) than expected by chance (randomized cell positions) at Δt = 0. **j** At Δt = 24 h, cluster overlap has further decreased. **k** Cluster overlap ($I_c$) drops with increasing Δt. Clusters of real cellular positions (top panel) overlap less than expected by chance (randomized cell positions, bottom panel). Source data are provided as a Source Data file.

day 6 ($\Delta t = 5$ days, Fig. 2a). In contrast to the $RT_{\Delta t = 1}$, cFos overlap in the $RT_{\Delta t = 5}$ was significantly lower and not different from overlap expected by chance (Fig. 5a–c, Supplementary Fig. 2). Thus, the cFos ensemble tagged in DG on the first day of WM training is not stable over multiple training days, but the neurons remain relevant for spatial memory retrieval (Figs. 3g–i, 4e).

We next explored the dynamics of cFos patterns in untrained mice, tagging cFos[+] neurons in DG and CA1 in the home cage. Two weeks after injecting both regions of TetTag mice with $AAV_9$-TRE-*mKate2*, mice were taken OFF Dox for a 24 h period and either immediately sacrificed, or given an i.p. Dox injection to rapidly close the tagging window and sacrificed after 6, 12, or 24 h (Fig. 5d). Overlap between cFos-tagged (mKate2[+]) and cFos[+] (shEGFP[+]) neurons was highest at the end of the OFF Dox window in both DG (59%) and CA1 (56%, Fig. 5e, f). During the next 24 h, the overlap dropped to 2% in DG but only to 27% in CA1, indicating different temporal dynamics of cFos expression in the two areas (Fig. 5e, f). In the DG the overlap at $\Delta$0h was 12-fold higher than expected due to chance and fell below-chance level within 24 h (Fig. 4g). CA1 cFos ensembles were larger (Supplementary Fig. 2), increasing the expected overlap due to chance. At $\Delta$0h overlap in CA1 was significantly higher than expected by chance but within 6 h was at chance levels (Fig. 4g). The very linear drop of ensemble similarity in DG was initially surprising (Fig. 4g), but could be readily reproduced in a numerical simulation of a negative feedback loop, including the below-chance overlap at $\Delta t = 24$ h (Fig. 4h). As the shEGFP reporter is more stable than cFos itself, endogenous cFos expression might change somewhat faster. These results indicate that in a stable environment and without training (i.e., home cage), the default mechanism is for DG GCs to shift cFos expression to a completely new ensemble every day.

## cFos-expressing GCs form spatial clusters that segregate over time

The sparse activity of GCs is controlled by robust inhibitory inputs from hilar interneurons[42]. We were interested in the spatial patterns of cFos-expressing GCs, which may not be randomly distributed, but could instead reflect the sphere of influence of individual interneurons. We used the maps generated in the home cage experiments (Fig. 5d) to analyze clusters of cFos[+] neurons at the different time points (Fig. 5i, j). As a measure of cluster overlap $I_c$, we divided the overlapping area by the union of both areas (see methods). While at $\Delta t = 0$, clusters were partially overlapping (high $I_c$), cluster overlap gradually decreased with time to very low values at $\Delta t = 24$ h (Fig. 5k). This result was not simply a consequence of the decreasing number of double-positive neurons because randomizing the positions of single- and double-positive neurons within the GC layer yielded clusters with higher overlap and less segregation over time. We conclude that the process of temporal segregation we describe here not only plays out in individual GCs, but also prevents the re-use of DG regions that were highly active 24 h ago.

## GCs possess an intrinsic cFos blocking mechanism

The decorrelation of cFos ensembles in DG could reflect changes in synaptic input (i.e. network-level effects), membrane excitability[43], or a cFos shut-off mechanism in individual GCs, e.g. by transcriptional or translational repression of cFos. A candidate suppressor of cFos transcription is $\Delta$FosB, a long-lived splice variant of FosB that accumulates in active neurons[26,44–46]. To find out whether $\Delta$FosB accumulates during WM training, we analyzed the DG of mice that were trained for 7 days in the WM, using a pan-FosB antibody that recognizes both FosB and $\Delta$FosB (pan-FosB, Fig. 6a). GCs had either high cFos expression or high pan-FosB, but rarely both, resulting in an inverse correlation that was highly significant in data pooled from 3 mice and in individual mice (Fig. 6b, Supplementary Fig. 8). When considering only GCs that were cFos-tagged on day 1 of WM training (magenta) we observed high

levels of pan-FosB on day 7 but low cFos (Fig. 6c, d). Using the same overlap analysis as in previous experiments, cFos overlap was significantly below chance in this $\Delta$6-day cohort (Fig. 6e) whereas overlap between day 1 cFos-tagged and day 7 pan-FosB[+] cells was 8-fold higher than expected by chance (Fig. 6f). Thus, cFos expression in GCs that previously expressed cFos in the WM appears to be inhibited, possibly by the FosB variant $\Delta$FosB. The high levels of pan-FosB staining suggest that indeed the cFos-tagged neurons were repeatedly activated during the subsequent WM training, giving ample time for $\Delta$FosB to accumulate in this specific population.

To test whether cFos suppression is a cell-autonomous process or whether it reflects a lack of synaptic input, we used a chemogenetic approach. We expressed the Gq-DREADD to directly activate neurons and trigger cFos expression in rat hippocampal slice cultures[47,48]. In DG, the first clozapine-N-oxide (CNO) application induced cFos in 59% of the DREADD-expressing GCs (Fig. 7a, Supplementary Fig. 9). A second CNO application 24 h later induced cFos in a much smaller fraction (17%). In CA1 pyramidal cells, conversely, the first and the second CNO application were equally effective in inducing cFos (Fig. 7b). To our surprise, we detected high levels of pan-FosB in CA1 neurons on the second day, which apparently did not interfere with their ability to express cFos. The pan-FosB AB detects both FosB and $\Delta$FosB, but only the latter splice variant is thought to suppress cFos[44]. Thus, while we could recapitulate GC-specific cFos inhibition in these slice experiments, we failed to specifically label the inhibiting agent.

To visualize $\Delta$FosB protein levels in individual neurons, we subtracted from the pan-FosB fluorescence the fluorescence signal from a second antibody that recognizes the C-terminus of FosB, which is absent in $\Delta$FosB (Supplementary Figure 10). In hippocampal slice cultures, we detected significantly more $\Delta$FosB in DG neurons than in CA1 neurons, in both stimulated and non-stimulated cultures. This area difference was even more pronounced in mice after 7 days of WM training, where $\Delta$FosB accumulated in many GCs, but not in CA1 pyramidal cells (Supplementary Figure 10).

## Spatial learning increases the number of GCs in a cFos-repressed state

If accumulated $\Delta$FosB inhibits cFos in GCs, concentrations of the two transcription factors should be inversely correlated. To test this, we compared home-caged mice with mice that were WM trained on two (spaced training) or on seven days (daily training). As anticipated, WM training increased the number of $\Delta$FosB[+] GC (Fig. 8a) as well as the average intensity of $\Delta$FosB[+] expression (Fig. 8b). In individual GCs, cFos expression was inversely correlated with $\Delta$FosB, and the strength of the correlation in individual mice increased with WM training (Fig. 8c). We generated kernel density estimates (KDE) of the actual data (Fig. 8d) and 100 KDEs of scrambled data to simulate the null hypothesis (no interaction between $\Delta$FosB and cFos). Compared to the expectation of the null hypothesis, GCs were overrepresented in two regions: Low cFos with high $\Delta$FosB intensity, which we termed the repressed state (Fig. 8e, upper left quadrant), and high cFos with low $\Delta$FosB, which we denote the permissive state (lower right quadrant). In home-cage mice, only 20% of analyzed GC were in the repressed state (Fig. 8f). The repressed fraction increased to 32% after spaced WM training and to 44% after daily training. Thus, training increased the number of cFos-repressed GCs, which was consistent across animals (see Supplementary Fig. 11 for more examples). Importantly, only GCs that expressed cFos or $\Delta$FosB were included in this analysis. GCs that expressed neither marker were presumably not very active in the WM. In the CA1 region of the same animals we rarely detected neurons with $\Delta$FosB signal, even after daily WM training (Supplementary Fig. 12). This suggests that regardless of training level, almost all CA1 pyramidal cells remained in the permissive state.

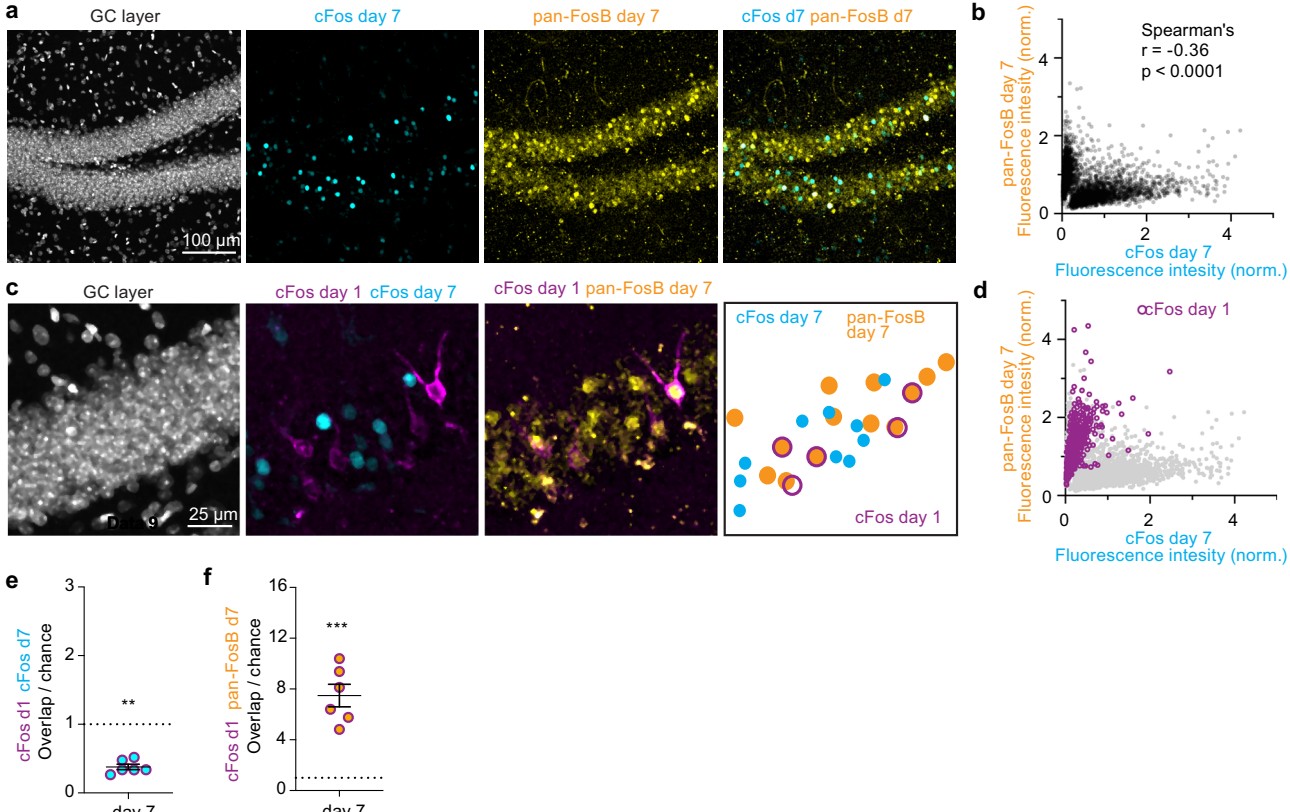

**Fig. 6 | Mechanism of cFos suppression in WM-trained mice. a** Immunostained hippocampal slice from the DG region against shEGFP (cFos, cyan) and pan-FosB mice (yellow) from mice that received daily training in the water maze for 7 days (from optogenetic experiments, Fig. 3). **b** Quantification of cFos and FosB/pan-FosB expression levels in individual granule cells (GC). GCs appear to be segregated in two clusters, resulting in a significant inverse correlation (****$p < 0.0001$, $n = 4111$ GCs from three mice, Spearman's $r = 0.36$). **c** Same tissue section as **a**, showing cFos$_{day1}$ (magenta), cFos$_{day7}$ (cyan), and pan-FosB$_{day7}$ (yellow) immunoreactive

cells. Drawing illustrates co-expression. **d** Day 1 cFos-tagged cells (magenta, pooled from three mice) frequently express pan-FosB. Gray points are replotted from **b**. **e** Overlap between cFos-tagged (magenta, day 1) and cFos (cyan, day 7) is significantly below-chance level (**$p = 0.002$, two-sided paired $t$ test, $n = 6$ mice, mean ± SEM). **f** Overlap between day 1 cFos$^+$ ensembles and FosB/ΔFosB on day 7 is significantly higher than chance (***$p = 0.0004$, two-sided paired $t$ test, $n = 6$ mice, mean ± SEM). Complete statistical information in Supplementary Table 1. Source data are provided as a Source Data file.

## Discussion

How time is represented in the brain is still a matter of debate. Using a multi-day training paradigm, we discovered a time-dependent shift in cFos expression patterns that seems to be specific to GCs of the DG. We observed that in individual GCs, accumulation of ΔFosB during water maze training was negatively correlated with cFos expression. Suppression of cFos transcription by ΔFosB has previously been described under pathological conditions (AD models, seizures)[44], and interfering with ΔFosB in hippocampal neurons is known to affect learning and memory[26]. Here we show that even in the home cage, ΔFosB suppression of cFos ensures that the cFos$^+$ ensemble of GCs changes from day to day. Given the extremely high stability of place cells in DG[10], we expected to find stable cFos$^+$ ensembles during multi-day training in a consistent environment, which was not the case. Two questions arise: What does cFos expression tell us about neuronal activity, and second, are cFos$^+$ GCs place cells?

While cFos is known to be driven by high-frequency spiking[27,49,50], the reverse is not true in the DG: The absence of cFos is not evidence for the absence of activity. This became clear when we inhibited the activity of the original (day 1) cFos-tagged ensemble on consecutive days: WM performance was compromised in trials with optogenetic inhibition, even though a very small fraction of GCs in dorsal DG (-1%) was silenced (Fig. 3). The inhibition of cFos-tagged neurons was specific, as inhibition of random GCs was much less effective: we observed behavioral effects only in animals where >30% of GCs were inhibited. Thus, GCs that expressed cFos on the first training day remained

particularly relevant for successful navigation in that environment, even though they rarely re-expressed cFos on the successive training days. We confirmed this by showing that cFos-tagged GCs are functionally reactivated when mice revisited the WM, but were not reactivated in a novel environment (Fig. 4). Therefore, low cFos overlap in DG is not in contradiction with the very high stability of place cells in DG[10]. Our findings do, however, indicate a problem when using cFos to identify engram cells. A defining criterion of engram cells stipulates that reactivation—presumably electric—in the same context must occur[17,19,51]. As cFos-tagged neurons in DG remain cFos$^-$ when electrically reactivated on subsequent days (Figs. 4 and 5), using cFos as a proxy for electrical activity is problematic. Defining only a few GCs that express cFos twice as the engram cells would strongly underestimate the number of neurons relevant for memory recall. The DG cFos$^+$ map produced on a given day will mostly contain GCs that have been activated for the first time in the WM. These cFos$^+$ GCs may encode new features being added to the internal representation, possibly acting as a timestamp on the most recently acquired memory. Our optogenetic inhibition experiments suggest that the full set of GCs required to recall the platform position includes many cFos-negative cells.

Why does electrical reactivation of GCs in a behavioral task rarely trigger renewed cFos expression? In slice cultures, we could reproduce this cFos refractory period by repeated chemogenetic stimulation of GCs. CA1 pyramidal cells, in contrast, readily express cFos on successive days. In search of the repressive mechanism, we turned to the FosB splice variant ΔFosB, which is known to be expressed in GCs and to

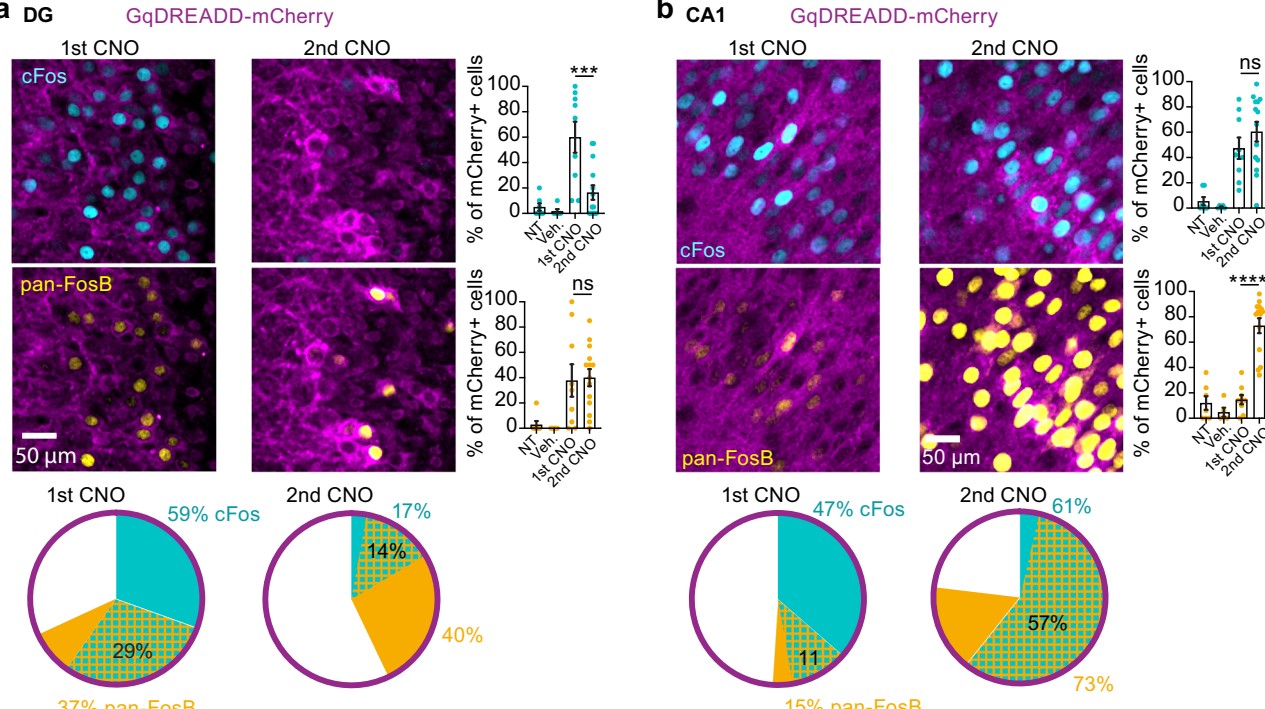

**Fig. 7 | Repeated chemogenetic induction of cFos expression in slice culture.**
**a** AAV transduction of hippocampal slice culture with CaMKaII-*hM3Dq-mCherry*
(magenta). Application of CNO activates DREADD and induces strong cFos (cyan)
expression in DG (1st CNO, *n* = 9 slices). A second application 24 h later triggered
significantly less cFos expression (***p = 0.0003, 2nd CNO, *n* = 14 slices, one-way-
ANOVA with Šídák's multiple comparisons test). No cFos was detected in cultures
with no treatment (NT, *n* = 7 slices) or treated with vehicle (Veh., *n* = 5 slices). On both
days, there was also pan-FosB immunoreactivity (yellow) which partially overlapped
with cFos (pie charts). **b** CA1 pyramidal cells respond to the second CNO stimulation
with renewed cFos expression in spite of high levels of pan-FosB. Bars show mean of
all slices ±SEM. Results were replicated at least three times. Complete statistical
information in Supplementary Table 1. Source data are provided as a Source Data file.

inhibit the expression of cFos in these neurons[26,44]. We quantified the
amount of ΔFosB in individual neurons, revealing the effect of WM
training on the GC population: With repeated training, more and more
GCs shifted from a low ΔFosB state that is permissive for cFos
expression to a high ΔFosB state, in which cFos is repressed. Behavioral
training did not have the same effect on CA1 neurons, the large
majority of which remained in the permissive state even after 7 days of
daily training. Interestingly, the few ΔFosB-positive CA1 neurons we
found were all cFos-negative, suggesting that the suppressive
mechanism is functional in pyramidal cells, but little ΔFosB is pro-
duced during behavior. In vivo imaging experiments have indeed
shown that in 60% of the cFos-expressing pyramidal neurons in CA1[47]
and 80% in barrel cortex[30], cFos expression is not transient, but sus-
tained for several days. We likewise observed that CA1 neurons express
cFos rather persistently, with 27% of homcage-tagged neurons con-
tinuing to express cFos 24 h after the mice were back on doxycycline,
compared to just 2% of GCs. Active suppression also explains the
perfectly linear drop in cFos levels over time we observed in GCs
(Fig. 5g) which could not be fitted by simple (exponential) decay
functions.

A new picture of time-controlled cFos regulation in the DG begins
to emerge. Only 19% of cFos-tagged GCs in the DG expressed cFos 12 h
later, and after 24 h, the overlap dropped below-chance level. When
neurons express cFos they are hyperexcitable[29,52,53], whereas over-
expression of ΔFosB reduces excitability[54]. Hyperexcitability increases
the likelihood of burst firing, creating ideal conditions for the poten-
tiation of incoming synapses[55]. Burst firing also maximizes the impact
of GCs on postsynaptic CA3 neurons, as the GC-CA3 mossy fiber
synapses display extremely strong short-term facilitation[56,57]. During
the high cFos period, the probability of long-term potentiation may
therefore be increased at both GC input and output synapses, i.e., onto
CA3 pyramidal cells and interneurons targeted by the current set of

cFos⁺ GCs[58]. In contrast, synaptic plasticity may be reduced during the
refractory period when ΔFosB is high, cFos expression is inhibited and
excitability is reduced, possibly write-protecting synapses on high
ΔFosB neurons from further modification. In nucleus accumbens, this
regulatory mechanism has been proposed to underlie the switch from
casual drug use to cocaine addiction[59]. In the amygdala, oscillations of
CREB/ICER have been suggested as a mechanism to group memories in
time[60].

The assumption that cFos is an indicator of highly active (engram)
neurons may hold for naive mice housed under standardized
(deprived) conditions and exposed to fear conditioning or other forms
of one-trial learning[61,62]. Multi-day training, which may be slightly clo-
ser to the daily challenges faced by mice in the wild, leads to the
accumulation of ΔFosB in many GCs and potentially other brain
areas[63]. Our results suggest that under these conditions, cFos patterns
in DG depend not only on input from the entorhinal cortex, but are
also gated by the activity history of each GC. Investigation of memory
systems "under load" during complex behaviors[64] may thus change our
view of hippocampal processing and engram formation.

## Methods
### Experimental animals

B6.Cg-Tg(Fos-tTA,Fos-EGFP*)1Mmay/J (TetTag) mice were obtained from the
Jackson Laboratory (Strain #018306) and bred to wildtype (noncarrier)
C57BL6/J mice from our colony. Mice were group-housed with litter-
mates until 2 weeks before rAAV injections, then were single-caged.
Mice had access to food and water *ad libitum* and were kept in an
animal facility next to the behavioral rooms on a reversed light-dark
cycle (dark 7 am–7 pm) at 20–23 °C with 45–65% humidity. All beha-
vioral experiments were done during the dark phase of the cycle. Due
to the requirement to swim with optical fibers, only male mice between
20–40 weeks (>28 g by the time of surgery) were used for optogenetic

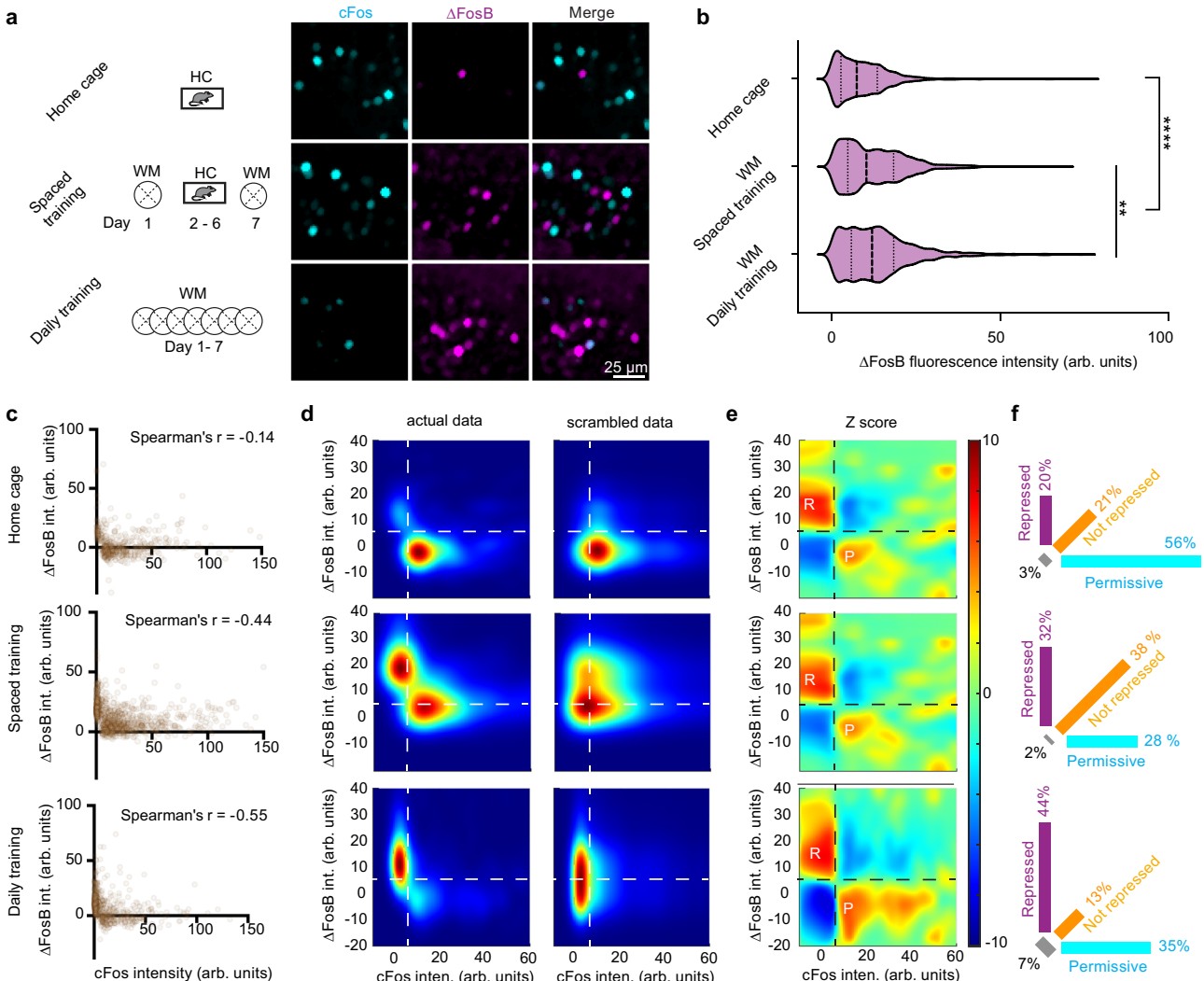

**Fig. 8 | Water maze training leads to accumulation of ΔFosB and cFos repression in DG. a** ΔFosB (magenta) and cFos expression (cyan) in DG was analyzed after 3 different training regimes: Home-caged (HC) mice, spaced training (ST) and daily training (DT). ΔFosB images show the difference in fluorescence between antibodies against the N-terminal and the C-terminal region of FosB (see Supplementary Figure 10). **b** Water maze training significantly increased ΔFosB in granule cells (GC) from DG (fluorescence intensity from 4 images per mouse. HC: 2 mice, $n = 514$ GCs; ST: 2 mice, $n = 1535$ GCs; DT: 3 mice, $n = 1344$ GCs). Violin plots showing a range of data distribution, median (dashed line), and quartiles (dotted lines). ΔFosB expression increases significantly with spatial learning (****$p < 0.0001$ HC vs ST/DT) ΔFosB expression is significantly higher in DT vs ST mice (**$p = 0.009$) Kruskal–Wallis with multiple comparisons Dunn test. **c** Single-cell analysis of

shEGFP (cFos) and ΔFosB expression in DG after the different training regimes (1 mouse each, see also Supplemental Fig. 11). WM training increased the strength of the inverse correlation between cFos and ΔFosB expression (Spearman's $r$). **d** Peak-scaled kernel density estimates of the data shown in **c**. Scrambled data is the average of 100 randomized combinations of the actual data. **e** Z-scores computed from **d**. Dashed lines are drawn between two regions of overrepresented cells (positive Z scores, red): Low cFos - high ΔFosB (R, repressed) and high cFos - low ΔFosB (P, permissive) and two underrepresented regions (blue). **f** Percentage of cells in the quadrants indicated in **d** and **e**. Magenta represents cells in the repressed state, orange in the not-repressed state, and cyan in the permissive state. Note that unlabeled GCs were not analyzed. Source data are provided as a Source Data file.

WM experiments (Figs. 3–5). Both male and female mice were included in the cFos ensemble overlap experiments (Figs. 1, 2 and 5). All experiments were conducted in accordance with German law and European Union directives on the protection of animals used for scientific purposes and were approved by the local authorities of the City of Hamburg (Behörde für Justiz und Verbraucherschutz, Lebensmittelsicherheit und Veterinärwesen, N 100/15 and N 046/2021).

### Viral constructs
To label the membrane of cFos[+] neurons, we used a red fluorescent protein fused to an opsin[65] (iChloC-linker-mKate2). This construct was inserted into a pAAV-TREtight backbone using MluI and EcoRI restriction enzymes to produce pAAV-TREtight-*iChloC-mKate2*. To drive spiking in cFos neurons, iChloC was replaced with CheRiff[66] to

create pAAV-TREtight-*CheRiff-mKate2*. To inhibit cFos[+] neurons, we used pAAV-TRE3G-*BiPOLES-mKate2* (Addgene # 192579). Constructs were packaged into AAV₉ (iChloC-mKate2) or into AAV_PHP.eB (BiPOLES, CheRiff) by the UKE Vector Facility. To photoconvert cFos[+] cells with high calcium levels, iChloC-mKate was replaced with CaMPARI2[40]. AAVs for the DREADD experiments were obtained from Addgene (AAV_AAV-CaMKIIa-*h4MD(Gi)-mCherry*, Addgene # 50476-AAV₉; AAV₉-CaMKIIa-*hM3Dq-mCherry*, Addgene # 50477-AAV₉).

### Stereotactic injection and fiber implant
TetTag mice were virus-injected under analgesia and anesthesia using a stereotaxic drill and injection robot (Neurostar). Mice were fixed to the frame under isoflurane anesthesia (1.5% mixed in $O_2$), skin and connective tissue was removed, and two craniotomies were performed

using an automated drill on the desired coordinates. AAVs were injected at $10^{12}$ vg/ml concentration, except for iChloC-mKate2 experiments where it was $10^{13}$ vg/ml. AAVs were delivered bilaterally into the dorsal hippocampus using a glass micropipette attached to a 5 µl syringe (Hamilton). A single injection per site was performed using stereotaxic coordinates for DG (−2.2 AP, ±1.37 ML, −1.9 DV) with a volume of 500 nl on each side (injection speed: 100 nl/min). CA1 was also injected in a set of experiments (Fig. 5) by moving to −1.4 DV and injecting 400 nl after the DG injection. After the last injection, the pipette was retracted 200 µm and left for at least 5 min to minimize efflux of virus during withdrawal. After the injections, the bone surface was cleaned with 0.9% NaCl solution and the skin was stitched. To avoid hypothermia, a heated pad was placed under the animal during surgery and under its cage for 1 h until full recovery. We provided post-surgery analgesia with Meloxicam mixed with softened Dox-food (see below) for 3 days after surgery. Animals recovered at least 2 weeks before behavioral experiments. Mice for optogenetic experiments were implanted with a custom-made bilateral tapered tip optic fiber implant (Doric) targeting the DG sulcus (−2.2 AP, ±1.37 ML, −1.7 DV) right after AAV injection. Implants were attached to the skull using dental cement (C&B Metabond). A protective cap made of an Eppendorf tube was secured by applying acrylic resin (Pattern Resin LS, GC America) to the exposed skull around the implant.

### Doxycycline treatment

Animals were given doxycycline-containing food (Altromin-Dox, 50 mg per kg of body weight, red pellets). To tag the first cFos ensemble, animals were changed to doxycycline-free food (Altromin, light-brown pellets) 24 h before exposure to the task. To ensure that the animals did not eat any Dox-food crumbles that had fallen into their cages, they were moved into new cages with fresh bedding. The old nesting material was transferred to the new cage to decrease novelty. Dox food was resupplied exactly 24 h after removal, right before the behavioral task. For cFos ensemble temporal shift experiments (Fig. 4), doxycycline (50 µg/g body weight) was injected I.P. after the end of the 24 h OFF Dox period.

### Behavioral experiments

For ensemble overlap experiments, mice were injected in batches and randomly assigned to an experimental group. Optogenetic experiments had a crossover design where all mice had probe trials with and without light, to minimize sampling errors and reduce the number of animals required to reach statistical power (paired statistics). Mice that did not have adequate viral transduction (see image analysis section), had off-target implants, or performed poorly (floaters or implant intolerance) were excluded from the analysis. All experiments were recorded on digital video. Ethovision XT 11.5 was used for automated tracking.

**Water Maze.** Animals were handled for 1 week before the start of the pre-training sessions to reduce stress during behavioral tasks. **Pre-training**: Mice were pre-trained for 2 days before their first exposure to the WM arena. Sessions (3-4 trials of max. 60 s each on two days) were done in a small rectangular water tank in the dark, in the same room where the WM task was performed. Water level was 1 cm above the 14 cm diameter escape platform. The position of the platform was alternated between the left and right side of the tank between trials, keeping a distance of 5 cm from the walls to avoid thigmotaxis. Once the animals found the platform, a grid was presented until the animals climbed onto it and were returned to their home cages in the waiting area of the behavioral room. **Training:** The WM consisted of a circular tank (1.45 m diameter) with visual asymmetrical landmarks, filled with water with (non-toxic) white paint. A platform (submerged by 1 cm) was placed in the center of the east quadrant during regular training and switched to the opposite (west) quadrant for reversal training

(max. swim time: 90 s). To test spatial reference memory, a probe trial (PT) without a platform was performed on each day. Mice underwent six trials every day (4 training trials (TTs, 90 s) + 1 PT (60 s) + 1 TT), inter-trial interval (ITI) was 8–10 s. For the TTs, mice were lowered into the tank facing the wall in different, pseudo-randomized positions (avoiding the target quadrant). In PTs, mice were lowered in the center of the tank. An opaque cup-sized chamber attached to a pole was used to transfer the mice from their home cage to the drop position and a plastic grid attached to a pole was used to pick up the mice. Mice were picked up 10 s after they found the platform and were returned to their home cage. Mice that did not find the platform during the TT were guided to it using the grid and were picked up after a 10 s on-platform waiting period. For both pre-training and WM, water temperature was 19–21 °C and a heat lamp was placed over the waiting area to prevent hypothermia.

**Home cage and open field control experiments.** To evaluate the temporal stability of cFos ensembles, mice were left unperturbed in their home cage (HC) during the OFF Dox period and sacrificed after 0, 6, 12, or 24 h ON-Dox. For all the cFos overlap experiments, both cFos$^+$ tagging events were designed to have the same amount of trials and training length. For the open field experiments, mice were placed inside a square arena (50 × 50 cm, 50 lux) for 20 min and stayed in the behavioral room at the same time as their WM counterparts.

**Optogenetic silencing in the water maze.** Mice were connected to two thin optic fibers (200 µm core diameter, 2 m, Doric Lenses, Canada) using ceramic ferrules (Doric) and put back in their home cage for at least 5 min before any behavioral task. In pre-training trials (2 days, three trials per day), mice were connected and disconnected before every trial to habituate them to the tether. To avoid stress-related cFos tagging, mice were never tethered during the Dox-OFF period. In all trials with tether, the weight of the fibers was compensated with a white helium balloon (0.08 N pull force), attached to a light fiber ~30 cm above the mouse with a transparent thin plastic tubing, long enough to keep the balloon out of the field of view of the video camera. Optogenetic silencing occurred only during memory recall (PTs, no target platform). Blue (473 nm) or yellow (594 nm) light was delivered using a laser combiner (LightHUB, Omicron) connected to a commutator (FRJ_1x2i_FC-2FC_0.22 Doric Lenses, Canada) to split the output into two fibers. The commutator was placed close to the ceiling, at the same height as the camera. Blue light delivery was started when mice were still in their HC (473 nm light pulsed at 33 Hz, 10 ms pulses, 8–10 mW). Immediately after light *on*, mice were placed on a starting platform facing the wall on the south quadrant. The starting platform was submerged with a hydraulic mechanism (Atlantis platform) after 30 s. After 60 s of swim time, the target platform (east quadrant) was raised to just below the water level. Mice that found the target platform was transferred to their HC 10 s later. In cases where mice did not find the platform they were guided to it and then transferred to their HC. Blue light was *on* for a total of 2 min, covering the entire WM trial. "No light" probe trials were done exactly the same, but with the blue laser *off*.

**Training protocol.** Two batches of mice were used for optogenetic manipulation in the water maze (Fig. 3). On training day 1, the protocol consisted of six trials: four TTs of max. 90 s with the platform in quadrant E, followed by a PT (60 s, without platform) and one more TT. On day 2, mice were placed in the starting Atlantis position for 90 s (visual recall trial). Afterwards, they performed two PTs while blue light was being delivered in the same crossover design, followed by three TTs. Day 3 started with two TTs, followed by two PTs (crossover design, blue light pulses), and two more TTs. On day 4 mice underwent the standard protocol (four TTs, one PT without light, one TT). Day 5 training, mice had two TTs followed by two PTs (blue light, crossover

design). We repeated this protocol on day 6 with yellow light pulses during PTs. This batch was perfused for immunohistochemistry on day 9. The second batch was trained similarly. After tagging (training day 1) mice went through the same training protocol from day 2 to day 4, adding a yellow light trial before the first TT. However, we found that yellow light had no effect on behavior, cFos expression, or cFos overlap. Consequently, we pooled data from "yellow light" and "no yellow light" animals to increase statistical power (Fig. 3h–j). On day 5, mice had two TTs, followed by two PTs, where half of the mice received blue light for the duration of the trial. On day 6, mice had two more TTs and a PT. This batch of mice was perfused for immunohistochemistry on day 7, 90 min after the last PT.

**Chemogenetic silencing in the WM**. Wild-type mice were bilaterally injected in DG with AAV9-*h4MD(Gi)-mCherry*. After at least 2 weeks of recovery, mice were handled and pre-trained like the previous groups. On day 1, mice were trained for 6 trials. On day 2, mice had two PTs. 40 min before the 1st PT, they received an intraperitoneal (I.P.) injection of 0.9% NaCl solution (vehicle). Mice were then injected with clozapine-N-oxide (CNO, 5 mg kg⁻¹) in 0.9% NaCl solution. After 40 min, their performance was tested again (2nd PT). From days 3–7, mice were trained every day with 4–6 TTs and 1 PT. On day 8, mice were injected again with CNO 40 min before the PT.

**CaMPARI2 conversion experiments**. TetTag mice were injected bilaterally with AAV$_{PHP.eB}$-TRE-*CaMPARI2* in DG ($1.2 \times 10^{12}$). Tapered fiber stubs were implanted bilaterally like in the optogenetic experiments. All mice were trained Off-Dox in the WM for 1 day. The following day (On-Dox), one group of mice were again trained in the WM. Photoconverting (PC) light (405 nm) was delivered throughout each WM trial in the ET group and the total illumination time was measured. For each WM-WM animal, we prepared a yoked control animal which, on the second day, received the same dose of PC light (same intensity, same duration) while exploring a novel environment (open field). 90 min after the last trial, mice were sacrificed for immunostaining (cFos-GFP, red CaMPARI2).

**Ex vivo brain processing and IF staining**
For cFos overlap analysis, mice were perfused 3 h after the last trial. In optogenetic silencing experiments, mice were perfused 1.5 h after the last trial. Mice were injected with ketamine/xylazine (100/10 mg kg⁻¹) intraperitoneal and intracardially perfused with 1x phosphate-buffered saline (PBS, Sigma) followed by 4% paraformaldehyde (PFA, Roth). Brains were extracted and stored in 4% PFA for at least 24 h at 4 °C. Before sectioning, brains were washed with 1× PBS for 20 min at room temperature (RT). The dorsal hippocampal region (−1.2 to −2.3 AP from bregma) was cut into 40–50 μm coronal sections using a vibratome (Leica VT100S) and collected in PBS. From each series, six sections were selected from Bregma −1.7 to −2.3 mm and incubated in blocking buffer (1× PBS, 0,3% TritonX, 5% goat serum) for 2 h at RT. Next, sections were placed in the primary antibody carrier solution (1× PBS, 0,3% TritonX, 1% goat serum, 1% BSA) and incubated overnight at 4 °C. After 3× washing with 1× PBS for 5 min, the sections were incubated for 2 h at RT in the secondary antibody carrier solution (1× PBS, 0.3% TritonX, 5% goat serum). Sections were washed 3 × 10 min in 1× PBS, stained with 4′,6-diamidino-2-phenylindole (DAPI, 1:1000) for 5 min and mounted on coverslips using Immu-Mount (Shandon). Complete antibody information in Supplementary Table 2.

**Confocal imaging**
Mounted brain slices were labeled with a code to blind the analyst. For cFos overlap experiments, immunostained slices were imaged with a confocal microscope (Olympus Fluoview FV 1000) using an oil immersion objective (UPLSAPO ×20/0.85). From each of the six slices per brain (left and right DG), 15 μm were imaged (stack of 10 images at

1024 × 1024 resolution, Z-step: 1.5 μm). Slices from BiPOLES-expressing animals were imaged with a Zeiss LSM 900 (Plan-Apochromat 20×/ 0.8). From each section, 18 μm were imaged (stack of six images at 1024 × 1024 resolution, Z-step: 3 μm). Excitation/emission filters were selected using the dye selection function of the Fluoview/ Zen3.5 software (Alexa 405 (DAPI), Alexa 488 (shEGFP) and Alexa 568 (mKate2), Alexa 647 (cFos or FosB). The image acquisition settings were optimized once and kept constant for all images within an experimental data set. Images were not deconvolved or filtered for quantitative analysis. Exemplary images presented in the figures are median-filtered (2 × 2 kernel) and cropped. Each color channel was linearly adjusted (Image J).

**Ensemble size calculation and overlap analysis**
Confocal image stacks (pseudo-colored cyan: shEGFP; pseudo-colored magenta: mKate2; pseudo-colored gray: DAPI) were analyzed with Imaris (Oxford Instruments). The volume of the upper and lower blade of DG (granule cell layer, GCL) based on the DAPI channel was used to estimate the total GC number based on published cell densities[67] and to mask the cyan and the magenta channel to restrict analysis to the GCL. Automatic spot detection was used to identify shEGFP⁺ cells in the cyan channel. The quality filter (round nuclei with a diameter of 8 μm) was adjusted once and then applied to every image stack of the same experiment. False positive spots (e.g., staining artifacts) were manually removed. It was not possible to detect mKate2-expressing GCs automatically as only the plasma membrane was labeled. To count mKate2-positive cells, spots were placed manually (spot size 12 μm) using the pseudo-colored magenta channel only. Double-positive cells (distance between shEGFP⁺ and mKate2⁺ spots <5 μm) were identified using a Matlab script. They were then manually inspected to check for artifacts and, if necessary, corrected. From the Imaris analysis, we calculated the following quantities:

(1) $\text{Number of granule cells} = \dfrac{\text{DAPI surface volume}}{\text{reported GC density}}$

(2) $\text{Fraction of shEGFP tagged cells} = \dfrac{\text{number of shEGFP + cells}}{\text{number of GCs}}$

(3) $\text{Fraction of mKate2 tagged cells} = \dfrac{\text{number of mKate2 + cells}}{\text{number of GCs}}$

(4) $\text{Fraction of double positive cells} = \dfrac{\text{number of double positive cells}}{\text{number of GCs}}$

(5) $\text{expected overlap} = \text{fraction of shEGFP cells} * \text{fraction of mKate2 cells}$

(6) $\text{Overlap/chance} = \dfrac{\text{fraction of double positive cells}}{\text{expected overlap}}$

**Cluster analysis**
Neurons were clustered by the Agglomerative Clustering algorithm implemented in the scikit-learn python library. For each image, we first computed the connection graph by using kneighbors_graph, which is also implemented in scikit-learn. The number of neighbors was fixed as one-tenth of the total number of neurons. First (mKate⁺) and second set (shEGFP⁺) of cFos-positive neurons were clustered separately (linkage criterion: "ward"). Clustering was performed for 16 different numbers of clusters $n_c \in [5, 20]$. To determine the optimal number of clusters ($n_c$) for each image, we generated 500 sets of artificial data from each image by randomly distributing the cFos⁺ neurons within the cell body layer of DG. Double-positive neurons were treated as a

third type, keeping their numbers identical to the experimental data. The cell body layer area was defined by the DAPI channel, and the size of each neuron was standardized (16*16 pixels). Each set of artificial data was clustered with the same procedure as the empirical data. For each clustering result, a clustering index

$$(7) \quad C_c = \left\langle \frac{\text{median}(D_{ij})_{j \in \text{cluster}}}{\min(D_{ij})_{j \notin \text{cluster}}} \right\rangle_{i \in U}$$

was calculated, where $D_{ij}$ is the distance between neurons $i$ and $j$. Smaller $c_c$ indicates stronger clustering. For each $n_c$, we calculated the proportion (of the 500 artificial sets) where $c_c$ was smaller in the artificial data than in the empirical data. The optimal $n_c$ was defined as the one with the smallest proportion, i.e., the strongest clustering compared to randomly distributed cells. In the case of multiple $n_c$ with the same smallest proportion, the largest $n_c$ was selected. The optimal $n_c$ was determined for mKate2$^+$ and shEGFP$^+$ clusters separately (always including the double-positive cells).

The sphere of influence of a cluster was defined as the convex hull of all its neurons, calculated by using ConvexHull implemented in SciPy python library. The overlap index between mKate2$^+$ clusters $M_i$ and shEGFP$^+$ clusters $S_j$ was defined as

$$(8) \quad I_c = (M_i \cap S_j)/(M_i \cup S_j).$$

In case of overlap between multiple clusters, the largest overlap was scored. Another 200 sets of artificial data were generated from each image and analyzed in an identical fashion to create cumulative distributions of $I_c$ values expected for randomly distributed cFos$^+$ cells (Fig. 5h).

### Analysis of CaMPARI2 photoconversion
The antibody combinations (primary/secondary) used were Anti-CaMPARI2-Red (4F61)/rabbit-568 and cFos/rat-647. Due to the high scattering of 405 nm light by the brain tissue, only slices where the implant site was visible were analyzed (200–250 µm from the implant side, Supplementary Fig. 7). The DAPI channel was used to create a surface of the GCL to estimate the total number of GCs. 12 µm spots (Imaris) were manually placed on converted (red) and non-converted (green) CaMPARI2$^+$ GCs. Spots were exported to Matlab, the fluorescence intensities of all three channels were extracted, calcium-dependent conversion of each CaMPARI2$^+$ cell was calculated (R/R +G) and correlated with its cFos intensity.

### Hippocampal slice culture
TetTag mice (male and female) or Wistar Unilever rats (Female, Envigo, HsdCpb:WU) were prepared at postnatal day 4–8[68]. Briefly, animals were anesthetized with 80% CO$_2$ 20% O$_2$ and decapitated. Hippocampi were dissected in cold dissection medium containing (in mM): 248 sucrose, 26 NaHCO$_3$, 10 glucose, 4 KCl, 5 MgCl$_2$, 1 CaCl$_2$, 2 kynurenic acid, 0.001% phenol red (310–320 mOsm kg$^{-1}$, saturated with 95% O$_2$, 5% CO$_2$, pH 7.4). Tissue was cut into 400 µM thick sections on a tissue chopper and cultured on membranes (Millipore PIC-MORG50) at 37 °C in 5% CO$_2$. No antibiotics were added to the slice culture medium which was partially exchanged (60–70%) twice per week and contained 394 ml Minimal Essential Medium (for 500 ml), 20% horse serum, 1 mM L-glutamine, 0.01 mg ml$^{-1}$ insulin, 1.45 ml 5 M NaCl, 2 mM MgSO$_4$, 1.44 mM CaCl$_2$, 0.00125% ascorbic acid, 13 mM D-glucose.

### cFos expression induced by chemogenetic stimulation
Organotypic rat slice cultures were injected with AAV9-CaMKIIa-*hM3Dq-mCherry* at DIV 14 using a Picospritzer III (Parker Hanna-fin). Virus was injected into DG, CA3, and CA1 (1.8 bar pulses, 50 ms duration). After 4–5 days of expression, slices were stimulated either once or twice by applying a 10 µl drop of CNO (1 µM, Tocris) in buffer (in mM: 145 NaCl, 10 HEPES, 25 D-glucose, 1.25 NaH$_2$PO$_4$, 2.5 KCl, 1 MgCl, 2 CaCl$_2$, 0.01 TTX, pH 7.4) on top of the slice 24 h apart. Culture inserts were transferred to a new well containing only the buffer for 3 min to wash off the CNO and then returned to the incubator in culture medium. In twice-stimulated cultures, the first stimulation was done without TTX to avoid homeostatic plasticity. As controls, we included slices that were not treated (NT group) or treated with TTX-containing buffer without CNO (vehicle). All slices were fixed 70 min after the last treatment and stained against cFos and pan-FosB. Analysis was performed blind. DREADD-expressing cells were randomly selected based on their mCherry signal (20 spots for DG, 50 spots for CA1 per slice), nuclear cFos and pan-FosB fluorescence intensity was extracted.

### Acute slice preparation and electrophysiology
Mice were decapitated under CO$_2$ anesthesia and the brains were dissected. Acute coronal slices (300 µm) were cut on a vibratome (Leica VT1000 S) in cold (4 °C) cutting solution containing (in mM): choline-chloride (110), KCl (2.5), NaH$_2$PO$_4$ (1.25), NaHCO$_3$ (25), MgCl$_2$ (7), CaCl$_2$ (0.5), glucose (25), sodium ascorbate (11.6), sodium pyruvate (3.1). Oxygenation and pH (7.4.) were maintained by bubbling (95% O$_2$, 5% CO$_2$). Slices were kept in a holding chamber with artificial cerebrospinal fluid (aCSF) containing (in mM): NaCl (125), KCl (2.5), NaH$_2$PO$_4$ (1.25), NaHCO$_3$ (26), MgCl$_2$ (1), CaCl$_2$ (2), glucose (10), saturated with 95% O$_2$, 5% CO$_2$. Brain slices recovered at 34 °C for at least 45 min before the start of whole-cell patch-clamp recordings (30−31 °C). Patch pipettes were pulled from borosilicate glass and had a resistance of 3–5 MΩ when filled with internal solution containing (in mM): K-gluconate (135), HEPES (10), MgCl$_2$ (4), Na$_2$-ATP (4), Na-GTP (0.4), Na$_2$-phosphocreatine (10), L-ascorbic acid (3), EGTA (0.2). Internal solution had pH 7.2 and 295 mOsm/L. The mKate2-positive cells in the DG were patched and signals acquired through an Axopatch 200B or Multiclamp 700B (Axon Instruments, Inc.), National Instruments A/D boards, and Matlab running Ephus software[69]. Action potential firing was electrically evoked by somatic current injection. Optogenetic stimulation was given through the objective (Olympus, ×60, 1.0 NA) at 473 or 594 nm (CoolLED, pE-4000). Data were analyzed with Matlab or Clampfit 10.7 (Molecular Devices).

### Statistical analysis
Numerical data from individual experiments were collected and ensemble sizes were calculated in MS Excel (version 16.0). Statistical tests were conducted in GraphPad Prism (version 8) and assumed an alpha level of 0.05. To analyze differences, we used one and two-way analysis of variance and the Šídák method to correct for multiple comparisons. In some cases, we used two-sided $t$ tests. A detailed description of experimental groups and statistical tests is provided in Supplementary Table 1. For all figures, *$P < 0.05$, **$P < 0.01$, ***$P < 0.001$, ****$P < 0.0001$.

### Reporting summary
Further information on research design is available in the Nature Research Reporting Summary linked to this article.

## Data availability
Source data are provided as a Source Data file. Other data are available upon request from the corresponding authors. Source data are provided with this paper.

## Code availability
Custom code to generate kernel density estimates in Matlab and to simulate cFos suppression is provided via GitHub (https://github.com/toertner/Kernel-density-estimate).

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

## Acknowledgements

We would like to thank Iris Ohmert, Jan Schröder, Katryn Sauter, Sabine Graf, Chantal-Joy Agbo, Tanja Stößner, and Karen Kesseler for excellent technical assistance. The pAAV-TRE3G-*BiPOLES-mKate2* construct (Addgene #192579) was a gift from J. Simon Wiegert. Ingke Braren from the UKE Vector Facility produced rAAV. We thank Mary Muhia, Irm Hermans-Borgmeyer, Silvia Rodriguez-Rozada, Brenna Fearey, and Sabine Hoffmeister-Ullerich for their discussions and helpful advice. This study was supported by the Deutsche Forschungsgemeinschaft (DFG) through Collaborative Research Center CRC 936 (#178316478) Projects B7 (F.M., T.G.O.), A2 (A.K.E.) and Z3 (C.C.H.), SFB TRR169-A2 (C.C.H., D.L.), Research Unit FOR 2419 (#278170285, MK; #282803474, CEG, TGO), and by Consejo Nacional para la Ciencia y Tecnología (CONACYT, Mexico, P.L.M).

## Author contributions

P.L.M., A.F., A.K.E., F.M., and T.G.O. conceived and designed experiments. P.L.M., A.F., L.B., and L.A. performed behavioral experiments. A.F., T.F., J.A., L.L., and L.B. performed electrophysiological experiments. D.L. and C.C.H. applied cluster analysis. A.F. and C.E.G. performed slice culture experiments. P.L.M., L.B., L.A., and F.M. analyzed behavioral data. Confocal imaging, immunohistochemistry, and cFos scoring were done by A.F. and L.B. P.L.M., M.K., A.K.E., C.C.H., F.M., and G.O. obtained funding and provided infrastructure. P.L.M., C.E.G., F.M., and T.G.O. wrote the paper. All authors edited and approved the manuscript.

## Funding

## Competing interests

The authors declare no competing interests.
