## [Peer Review File · Nature Communications]

Δ FosB accumulation in hippocampal granule cells drives cFos pattern separation during spatial learningReviewers' comments:

Reviewer #1 (Remarks to the Author):

This exciting manuscript by Paul J. Lamothe-Molina, Thomas G. Oertner, and colleagues uses cutting-edge viral, transgenic, and clever behavioral techniques to tease out how hippocampus-mediated memories segregate over time, specifically at the level of tagged “engrams” in the dentate gyrus (DG). The authors findings are in abundance and highly influential: they provide a technically novel approach to visualizing two discrete snapshots of experience by combining a transgenic cFos mouse line with a virus-based c-Fos-driven reporter; daily training in the Morris Water Maze (MWM) task recruits differential sets of DG cells each day; optogenetically suppressing the activity of the set of cells tagged during initial training actually improved subsequent reversal learning in the MWM, which in and of itself is an impressive and difficult array of experiments to achieve; and, tagged DG cells partially overlap during initial training but subsequently segregate as animals experience a novel environment. These data are far-ranging and greatly enhance our understanding of the basic mechanisms of learning and memory, while also providing the field with an exciting set of discoveries with regards to how engrams evolve over time in general.

Moreover, while this report provides a novel neuronal and behavioral framework for interrogating the cellular evolution of learning and memory, I have a few minor points that, if addressed (or some least discussed), will elevate this manuscript further into a well-controlled, meticulous, and systematic dissection of modulating engrams, which collectively would make the manuscript very suitable for publication in Nature Communications. I am delighted to sign off on this review as Steve Ramirez (Assistant professor of psychological and brain sciences at Boston University) and fully support the current manuscript’s impact and integrity.

Minor points:

1. I’m curious if the authors think that the segregation here observed is a function of cFos activity in particular and whether or not this would extend to other activity-dependent genes (i.e. NPAS4 and cFos in Sun et al. Cell 2020); while any experiments here truly would be outside the scope of the manuscript, perhaps an expanded few sentences in the discussion could nuance the varieties of IEGs and their expression kinetics in the DG over time.
2. In the discussion, while the authors briefly touch on the subject of IEG-related activity and its ~hours-length timescale in relationship to both in vivo physiology’s ~ms to minutes timescales as well as putative potentiation mechanisms involved, I would be thrilled to hear what the authors think the DG in general is doing when considering the following: previous data shows that the DG undergoes rate remapping across tens of minutes when an environment is “morphed” (e.g. Leutgeb et al. 2007), which is nicely consistent with the partial overlap observed in early training in the present manuscript, while IEG-based studies show global remapping / “separation” occurring as populations are orthogonalized over the course of days. In other words, environmental changes occurring close in time seem to promote in the DG a kind of partial overlap in IEG expression as well as rate remapping using in vivo physiology (could they be related?), and a kind of global remapping occurring over days is evidenced by two different populations representing each experience (based on IEG and in vivo physiology results). Do the authors speculate that any two experience, regardless of how different or similar and regardless of spatial or emotional components, would partially recruit the same population of cells if they occur close in time and then drift to two separate populations over time (e.g. Figure 4E’s data)? Indeed, the “temporal shift” mentioned in line 363 is tantalizing and appears in many brain areas, as noted, and a slightly expanded discussion on the advantages of having ensembles undergo temporal shift would nicely balance out the end of the paper.
- 3) When silencing the original encoding ensemble, the authors strikingly observed an improvement in recall of the first platform as well as an improvement in reversal learning. While certainly outside the experimental scope of the manuscript, I’m curious if the authors think the converse would be true: namely, that activating the original encoding ensemble would lead to a deficit in recall and an impairment in reversal learning?

Minor Stylistic Comments:

- In Figure 1, the panel labels and figure legends sometimes do not match
- In Figure 1, panel e, expressing the vertical axis title as a "Percentage Positive cells / DAPI" or "(%) Positive cells / DAPI" may enhance clarity
- In Figure 2, panel b, the white lettering for "He," is low contrast against the grey balloon. Black lettering would improve visibility
- Figure 2, panels d and e appear to be switched (d should be where e is, possibly?)
- Figure 4B: the representative histology images are out of alignment
- Figure 5: the panel labels and figure legend do not match
- Labeling figure 5, panel b's vertical axis as "latency (s)" would enhance consistency with other graphs.
- Figure 6: the panel labels and figure legend do not match
- The parentheses are not closed in the description for figure 6, panel b
- GC is introduced as an abbreviation in line 11. Please spell out "granule cells" before introducing GC as an abbreviation to avoid confusion.
- Line 273: hyperpolarizing (typo)
- References: an extra colon appears in the 29th reference (typo)
- Lines 610-611: the parentheses are not closed (typo)
- There are 2 extra colons in line 475 of the methodology (experimental animals subheading / page 31)

Overall, I very much applaud the author's manuscript and their efforts as well on their series of discoveries and their ambitious approach to these outstanding questions on hippocampal engrams—I very much look forward to their response and would like to reiterate this paper's suitability for publication in Nature Communications.

Reviewer #2 (Remarks to the Author):

Lamothe-Molina et al. investigate possible roles of ensembles of cFos-expressing hippocampal dentate gyrus (DG) neurons (mostly granule cells) in spatial tasks. They use an inducible TetTag strategy to access cFos-expressing neurons in a temporally defined manner ("permanent label"), and compare these neurons to those expressing cFos in a non-conditional manner ("recent label" due to short half-life of GFP construct). When mice learn the water maze (protocol consisting of 5 training trials and one probe trial), silencing a day1-ensemble leads to better reversal learning in expert mice (effect mainly detectable on the 4th of 4 trials during reversal learning), and to a small improvement in the maze learning curve in trained (day5) mice, but not in learning (day3) mice. The authors conclude from these somewhat unexpected findings that activity in DG neurons might not reflect spatial position and spatial memory. They then show that ensemble comparisons at different time points in the home cage reveal essentially no overlap at 24h time difference. Ensemble overlaps from day1 to day2 of water maze learning were very low but above chance, whereas they were not above chance if mice explored a novel environment on day2. Overlaps between day5 and day6 DG neuronal ensembles were not higher than chance, whereas the overlap was (slightly) above chance if mice were subjected to a reversal trial (platform in the opposite position) on day6. Overlaps determined with this method in DG neurons are thus generally low. They are slightly above chance when mice are learning about space, and particularly low (below chance) when mice are familiar with a particular setting (home cage or water maze experts).

The authors conclude that ensembles of cFos-expressing ("i.e. active") DG neurons reflect neither spatial position nor spatial memory. The authors further suggest that their findings are consistent with the notion that ensembles of "active neurons" in DG (and possibly in other neuronal subpopulations as well) might reflect a time-stamping mechanism for hippocampal memories.

The analysis of DG cFos ensemble overlaps using methods differing from those of most previous

studies, and the combination with optogenetic silencing is in principle of interest. However, while contradicting findings from several previous studies (not really clear why), this particular study does not provide firm novel conclusions, or conclusive novel insights as to the role of DG ensembles. The conclusion about position coding seems compelling, but the implications for hippocampal memories, and in particular for the contributions of the DG, are unclear. Several previous studies have provided evidence that the role of the DG in hippocampal function likely involves diversification (pattern separation). This particular study tends to confirm this view without adding novel insights. Overall, while this study seems to contradict several previous ones, the reasons for the discrepancies are unclear, and the study does not provide new insights as to the function of the DG or of cFos-expressing ensembles. As such, its significance is unclear.

Further points:

1) Spatial coding and in particular spatial memory likely involve hippocampal CA1 (and to some extent CA3). While the role of the DG has remained less clear, there is little to suggest that it contributes to spatial memory. If the purpose of the study was to investigate how cFos ensembles in the hippocampus reflect position and spatial memory, it would seem that CA1 would have been a much more plausible place to start, particularly when using new methods.

2) The hippocampus is thought to provide cognitive maps of subjective experiences, and spatial coding is just one aspect of such maps. It is therefore not a priori clear that cFos ensembles likely reflecting memory should necessarily involve some explicit representation of space.

3) The effects of the ensemble silencing experiments are very small, which renders the interpretation of these unexpected findings difficult, particularly in view of current uncertainties as to the role of DG ensembles.

4) The authors use terms such as "active neurons" and cFos-expressing neurons as if they were interchangeable. There is no question that the expression of cFos depends on neuronal activity, but there is little reason to believe that ensembles of cFos expressing neurons exactly reflect ensembles of active neurons.

Reviewer #3 (Remarks to the Author):

The manuscript by Lamothe-Molina et al characterized how active ensembles in the dentate gyrus of mice change over time, even when mice are exposed to a constant environment or subjected to the same spatial navigation task over days. The authors used TetTag mice expressing a short half-life GFP under a cFos promoter to transiently tag neurons that were active within hours prior to histological analysis, along with viral injection of an activity-dependent inhibitory opsin (iChloc) to permanently tag and manipulate neurons that were active during a 24h window when mice were taken off a doxycycline diet. Permanent tagging with iChloc occurred in the home-cage or with water maze training.

Overall, the authors' data shows relatively little overlap between active neurons permanently tagged during spatial learning and neurons active during further training or testing 1d or more later.

Furthermore, optogenetic inhibition of neurons that were tagged during early learning did not impede performance when mice were later tested for spatial memory. These results suggest that cFos ensembles in the dentate gyrus change over time and do not represent a map of space.

While there is an abundance of intriguing data in this study and the conclusions are in line with another recent study by Tanaka et al (2018) in the CA1 region of hippocampus, I have a number of major and minor comments that should be addressed.

Major comments:

1. In Figure 1 the equivalent levels of GFP expression in HC (0d, 1d), as well as in the WM groups (0d, 1d) shows that GFP reflects background/baseline fluctuations in fos expression/neuronal activity (which is further confirmed in Figure 4). It is compelling to think that this population may reflect

neurons within a given time frame from which ensemble/engram neurons can be chosen to encode the memory – supported by the fact that iChloc expression does not rise above GFP levels after WM training. However, as most studies in the DG (using fos, arc immunolabeling or other methods) show anywhere from 1-5% of cells become engram neurons, it means that anywhere from 50-90% of fos+ (or even iChloc+) cells are not part of the engram. This of course would have an impact on interpretation of the overlap calculations.

It would be critical, in my view, to be able to identify true engram neurons for overlap analysis, possibly by differences in fluorescence (as alluded to later in the manuscript). In addition, it is possible that the immunofluorescent amplification of signals masked any overt differences between low fos+ (non-engram) and high fos+ (engram) neurons.

2. The optogenetic silencing experiment in Figure 2 is surprising and interesting. However, there is no certainty that iChloc is sufficiently tagging the relevant ensemble since overlap may not be more than 20% as suggested by Figure 4 (and some of these iChloc+ cells may be those tagged under basal conditions rather than WM training). Also, is 1 day of training (when mice haven't sufficiently learned) able to define the final memory ensemble on Day 3 or Day 5? I don't believe it is known if other neurons become fos+ as training proceeds on the second and third day etc to add to the final memory ensemble – if so, that would suggest that optogenetic inhibition only inhibits a fraction of the memory ensemble on Day 5 and this is not due to time-dependent drift.

3. It would also be a logical progression to perform the optogenetic inhibition on WM early training (ET) mice used in Figure 5, comparing d1 vs d2, as well as WM overtraining (OT) mice, comparing d5 and d6. As d5/d6 mice show surprisingly little overlap, possibly for reasons presented in the Discussion, it would also be useful to do a similar experiment comparing d2 and d3, d3 and d4, d4 and d5 (i.e. iChloc tag on d4, inhibit during training on d5).

4. The above comment additionally highlights a need for a positive control for optogenetic inhibition of behavior. Contextual fear conditioning would be ideal as this has been previously shown.

5. Figure 4e is the most convincing evidence for shGFP turnover within hours, but there are not enough data points. I suggest adding 12h and 18h groups. It feels like wishful thinking to describe this as a sigmoidal function based on 3 data points.

6. The in vitro data showing shGFP turnover (Extended Data Figure 1d) is not completely clear to me. As far as I understand it, whole-image fluorescence peaks 3h after stimulation and returns to almost baseline by 6h. Does that leave open the possibility that there is still low fluorescence that can be detected? This figure also shows that fluorescence in "spots" peaks at 3h and stays level. What are spots? The Methods section seem to imply the diameter of these spots correspond to cells – if that is the case, then number of fluorescent cells doesn't go down. How does this fit with the whole image fluorescence? I also note that one graph goes to 5h while the other goes to 6h on the x-axis.

7. In a number of experiments, the number of mice used is low (e.g. Figure 1e: 3 mice in some groups, 7 mice in others; Figure 6d: n=4 and 5). This reduces the statistical power of these experiments.

8. According to the Methods, the mice used were 20-60 weeks of age (3.5-14 months, approximately). This is a huge variation and there have been studies that show alterations in hippocampal synaptic plasticity and memory in aged mice (>1yr). As such, it would be incumbent for the authors to examine the behaviour and fos/iChloc overlap in 20 week old vs 60 week old mice and show no difference, or account for any differences when performing their analysis.

9. Statistical analyses. Important details of statistical analyses are missing. For example, the data in Fig. 3e were analysed by mixed 2 way ANOVA, but main effects and interactions are not reported. Post-hocs (such as the Tukey post hoc test reported here) should only be reported following a significant interaction term. This same criticism applies the data in 3f, 5c.

10. The argument that these data provide evidence that the DG is doing some form of temporal pattern separation or time stamping is interesting. However, this might also be true of any region in the brain. For example, would the authors expect the same results were they to look at other regions processing spatial memory information such as CA1, retrosplenial cortex and so on. The current arguments would be greatly strengthened were the authors able to show that this is only happening in the DG.

Minor comments:

9. The schematic of shGFP expression in all the figures is somewhat misleading – it suggests a

specific population shows gradually increased GFP expression which then reduces to 0 over 6h, and then another population takes over (and thus there is a period with no or low GFP expression), rather than neurons that become fos+ and other neurons that become fos- over constant time periods. I realize the schematic is to illustrate a larger idea and so perhaps the reality should at least be acknowledged.

10. In Figure 1b, it appears that different high magnification images were used for mKate2, shGFP and overlap – it would be better to use one image in the different channels to show the sort of expression one expects.

11. What is “adequate expression” for the inclusion/exclusion criteria of virus infection (Methods, p33)? This should be detailed further.

12. I’m not sure prototypical fear conditioning is primarily considered a model for PTSD as suggested on p23 of the Discussion.

13. There are some typos, formatting errors and at least one mismatch between papers cited in the main text and the References section.

NCOMMS-20-36416A-Z, appeal / major revision

We would like to thank the reviewers for their helpful suggestions and guidance. In response, we have repeated all behavioral experiments with a more powerful inhibitory tool and repeated the cFos overlap analysis with a more stringent threshold. Please find our point-by-point response below.

Reviewer #1 (Remarks to the Author):

This exciting manuscript by Paul J. Lamothe-Molina, Thomas G. Oertner, and colleagues uses cutting-edge viral, transgenic, and clever behavioral techniques to tease out how hippocampus-mediated memories segregate over time, specifically at the level of tagged “engrams” in the dentate gyrus (DG). The authors findings are in abundance and highly influential: they provide a technically novel approach to visualizing two discrete snapshots of experience by combining a transgenic cFos mouse line with a virus-based c-Fos-driven reporter; daily training in the Morris Water Maze (MWM) task recruits differential sets of DG cells each day; optogenetically suppressing the activity of the set of cells tagged during initial training actually improved subsequent reversal learning in the MWM, which in and of itself is an impressive and difficult array of experiments to achieve; and, tagged DG cells partially overlap during initial training but subsequently segregate as animals experience a novel environment. These data are far-ranging and greatly enhance our understanding of the basic mechanisms of learning and memory, while also providing the field with an exciting set of discoveries with regards to how engrams evolve over time in general.

Moreover, while this report provides a novel neuronal and behavioral framework for interrogating the cellular evolution of learning and memory, I have a few minor points that, if addressed (or some least discussed), will elevate this manuscript further into a well-controlled, meticulous, and systematic dissection of modulating engrams, which collectively would make the manuscript very suitable for publication in Nature Communications. I am delighted to sign off on this review as Steve Ramirez (Assistant professor of psychological and brain sciences at Boston University) and fully support the current manuscript’s impact and integrity.

Minor points:

1. I’m curious if the authors think that the segregation here observed is a function of cFos activity in particular and whether or not this would extend to other activity-dependent genes (i.e. NPAS4 and cFos in Sun et al. Cell 2020); while any experiments here truly would be outside the scope of the manuscript, perhaps an expanded few sentences in the discussion could nuance the varieties of IEGs and their expression kinetics in the DG over time.

It would be very interesting to compare cFos to other IEG, but it would involve repeating all behavioral experiments in different mouse line (Arc-TRAP) or use the 1-viral approach RAM promoter used in Sun et al. 2020. Lacagnina et al. (2019) observed that the temporal stability of Arc-tagged cells is stable as long as there is no fear extinction. Interestingly, when mice visit the context (US) without shock (CS) to extinct the fear memory, a new Arc-ensemble emerges. This was not the case in our WM experiments, and it is difficult to know at this point if this is due to the nature of the IEG or the nature of the behavioral task. We now explicitly mention this discrepancy (lines 227-229).

2. In the discussion, while the authors briefly touch on the subject of IEG-related activity and it’s ~hours-length timescale in relationship to both in vivo physiology’s ~ms to minutes timescales as well as putative potentiation mechanisms involved, I would be thrilled to hear what the authors think the DG in general is doing when considering the following: previous data shows that the DG undergoes

rate remapping across tens of minutes when an environment is “morphed” (e.g. Leutgeb et al. 2007), which is nicely consistent with the partial overlap observed in early training in the present manuscript, while IEG-based studies show global remapping / “separation” occurring as populations are orthogonalized over the course of days. In other words, environmental changes occurring close in time seem to promote in the DG a kind of partial overlap in IEG expression as well as rate remapping using in vivo physiology (could they be related?), and a kind of global remapping occurring over days is evidenced by two different populations representing each experience (based on IEG and in vivo physiology results). Do the authors speculate that any two experience, regardless of how different or similar and regardless of spatial or emotional components, would partially recruit the same population of cells if they occur close in time and then drift to two separate populations over time (e.g. Figure 4E’s data)? Indeed, the “temporal shift” mentioned in line 363 is tantalizing and appears in many brain areas, as noted, and a slightly expanded discussion on the advantages of having ensembles undergo temporal shift would nicely balance out the end of the paper.

We have rewritten the discussion, based on a new set of optogenetic inhibition experiments that revealed a clear performance decrease when inhibiting the day 1 cFos ensemble. Together with the overlap data, we now conclude that is incorrect to equate lack of cFos reactivation with lack of activity during memory recall. We also discuss a potential mechanism for the time shift in DG: Accumulation of Δ fosB after several training days, a known inhibitor of cFos. Our results do not contradict the previous finding of stable place cells in DG (Hainmüller & Bartos), as these place-specific granule cells would not be able to express cFos again on the following days. While the time-dependent cFos pattern migration in DG fits very well to what we would expect from an encoder of episodic memories, we have little to say about the remapping of place cells in different environments. In the revised discussion, we hopefully explain our epistemic status regarding these questions more clearly.

3) When silencing the original encoding ensemble, the authors strikingly observed an improvement in recall of the first platform as well as an improvement in reversal learning. While certainly outside the experimental scope of the manuscript, I’m curious if the authors think the converse would be true: namely, that activating the original encoding ensemble would lead to a deficit in recall and an impairment in reversal learning?

We were not happy with the highly variable results produced by iChloC activation. We therefore repeated the optogenetic silencing experiments with several methodological improvements (better chloride channel, better promotor, Atlantis platform, see below) and found that spatial memory recall was consistently *impaired* when the day-1 cFos ensemble was silenced on day 3 or day 5. The new results are in line with experiments testing other types of memory (e.g. fear conditioning).

Inspired by this comment, we also performed experiments in which we drove spiking in the day-1 cFos ensemble using Chrimson (part of the BiPOLES construct). However, we found no effect on memory recall when we re-activated the cFos_{day1} ensemble during probe trials on day 6 (Supplementary Fig. 6). We know that these cells must be active during navigation (as silencing them decreases performance), so perhaps inducing additional spikes is not too disruptive. A negative result is always difficult to interpret, especially since blue light pulses are strongly scattered and reach only a small region of DG.

Minor Stylistic Comments:

- In Figure 1, the panel labels and figure legends sometimes do not match
- In Figure 1, panel e, expressing the vertical axis title as a “Percentage Positive cells / DAPI” or “(% Positive cells / DAPI” may enhance clarity
- In Figure 2, panel b, the white lettering for “He,” is low contrast against the grey balloon. Black lettering would improve visibility
- Figure 2, panels d and e appear to be switched (d should be where e is, possibly?)
- Figure 4B: the representative histology images are out of alignment
- Figure 5: the panel labels and figure legend do not match
- Labeling figure 5, panel b’s vertical axis as “latency (s)” would enhance consistency with other graphs.
- Figure 6: the panel labels and figure legend do not match
- The parentheses are not closed in the description for figure 6, panel b
- GC is introduced as an abbreviation in line 11. Please spell out “granule cells” before introducing GC as an abbreviation to avoid confusion.
- Line 273: hyperpolarizing (typo)
- References: an extra colon appears in the 29th reference (typo)
- Lines 610-611: the parentheses are not closed (typo)
- There are 2 extra colons in line 475 of the methodology (experimental animals subheading / page 31)

We have corrected all these typos and mismatches.

Overall, I very much applaud the author’s manuscript and their efforts as well on their series of discoveries and their ambitious approach to these outstanding questions on hippocampal engrams—I very much look forward to their response and would like to reiterate this paper’s suitability for publication in Nature Communications.

Reviewer #2 (Remarks to the Author):

Lamothe-Molina et al. investigate possible roles of ensembles of cFos-expressing hippocampal dentate gyrus (DG) neurons (mostly granule cells) in spatial tasks. They use an inducible TetTag strategy to access cFos-expressing neurons in a temporally defined manner (“permanent label”), and compare these neurons to those expressing cFos in a non-conditional manner (“recent label” due to short half-life of GFP construct). When mice learn the water maze (protocol consisting of 5 training trials and one probe trial), silencing a day1-ensemble leads to better reversal learning in expert mice (effect mainly detectable on the 4th of 4 trials during reversal learning), and to a small improvement in the maze learning curve in trained (day5) mice, but not in learning (day3) mice. The authors conclude from these somewhat unexpected findings that activity in DG neurons might not reflect spatial position and spatial memory. They then show that ensemble comparisons at different time points in the home cage reveal essentially no overlap at 24h time difference. Ensemble overlaps from day1 to day2 of water maze learning were very low but above chance, whereas they were not above chance if mice explored a novel environment on day2. Overlaps between day5 and day6 DG neuronal ensembles were not higher than chance, whereas the overlap was (slightly) above chance if mice were subjected to a reversal trial (platform in the opposite position) on day6. Overlaps determined with this method in DG neurons are thus generally low. They are slightly above chance when mice

are learning about space, and particularly low (below chance) when mice are familiar with a particular setting (home cage or water maze experts).

The authors conclude that ensembles of cFos-expressing ("i.e. active") DG neurons reflect neither spatial position nor spatial memory. The authors further suggest that their findings are consistent with the notion that ensembles of "active neurons" in DG (and possibly in other neuronal subpopulations as well) might reflect a time-stamping mechanism for hippocampal memories.

The analysis of DG cFos ensemble overlaps using methods differing from those of most previous studies, and the combination with optogenetic silencing is in principle of interest. However, while contradicting findings from several previous studies (not really clear why), ...

Worried about the discrepancies to other studies and the small and variable effect seen in the optogenetic silencing experiments using iChloC, we decided to repeat the optogenetic manipulation with improved methodology:

- Inhibitory opsin with higher conductance. We changed iChloC for the chloride-conducting rhodopsin (GtACR2) which produces more than 5 times higher photocurrents than iChloC with the same 473 nm light power (REF). This increases the tissue volume in which spiking is reliably inhibited.
- We used a soma-targeting sequence to reduce the undesired effects Cl⁻ influx in axonal and dendritic compartments. This way we ensure Cl⁻ influx leads to shunting inhibition of action potentials.
- We used tapered fiber implants instead of blunt fibers to further increase the irradiated volume and decrease mechanical tissue damage.
- We changed the tTA-response element promoter from TREtight to TRE3G. This reduced unspecific expression of the optogenetic tool (Supplementary Fig. 3).
- Modified water maze: We implemented a hydraulic system to place the mice on a starting platform which is submerged when desired (Atlantis platform). This reduced stress, improved performance and in addition, we had fewer tangles of the optic fibers with the helium balloon cord.

After repeating the silencing experiments with these improved methods, we observed that memory recall was impaired in every single animal, consistent with published inhibition studies on other forms of memory (e.g. fear conditioning).

... this particular study does not provide firm novel conclusions, or conclusive novel insights as to the role of DG ensembles. The conclusion about position coding seems compelling, but the implications for hippocampal memories, and in particular for the contributions of the DG, are unclear.

We have added new experiments in organotypic slice culture and in vivo, showing strong accumulation of Δ fosB in previously active granule cells during water maze training. This long-lived splice variant can inhibit cFos expression for several days. Furthermore, using optogenetic inhibition, we demonstrate that the day 1 cFos ensemble continues to be important for successful navigation, even though the large majority of these neurons do not express cFos again in the following days.

We hope the reviewer agrees that the daily shift in cFos expression, which we are starting to understand on the mechanistic level, may very well be a key feature for the formation of episodic memories. It also shows that cFos does not necessarily label the acutely most active cells, but is controlled by a cell-autonomous record of activity that stretches back several days. For future investigations of the Engram, these are important and novel insights.

Several previous studies have provided evidence that the role of the DG in hippocampal function likely involves diversification (pattern separation). This particular study tends to confirm this view without adding novel insights. Overall, while this study seems to contradict several previous ones, the reasons for the discrepancies are unclear, and the study does not provide new insights as to the function of the DG or of cFos-expressing ensembles. As such, its significance is unclear.

Our new set of optogenetic inhibition experiments shows that the original cFos ensemble continues to be important for solving the task during the following days. This is consistent with optogenetic and chemogenetic studies using fear conditioning, resolving the conflict of the previous manuscript. The significance of our study is expanded by presenting a mechanism for the temporal shift in cFos expression that is specific to DG.

Further points:

1) Spatial coding and in particular spatial memory likely involve hippocampal CA1 (and to some extent CA3). While the role of the DG has remained less clear, there is little to suggest that it contributes to spatial memory. If the purpose of the study was to investigate how cFos ensembles in the hippocampus reflect position and spatial memory, it would seem that CA1 would have been a much more plausible place to start, particularly when using new methods.

Preliminary data from the lab of Fabio Morellini showed that cFos is elevated during reversal learning in DG, but not in CA1. Moreover, the high stability of place cells in DG (Hainmüller & Bartos) led us to think we could manipulate place-encoding cells over the course of training (6 days). We certainly agree with the statement that ‘the role of the DG [in spatial coding] has remained less clear’, and we think our study shines new light on some of the confusion surrounding granule cell function. For starters, it is a big mistake to take cFos expression as a criterion for high activity, as it is highly regulated on the transcriptional level. We show Δ fosB accumulation in DG during multi-day training, but much less in CA1.

We have to admit, though, that our optogenetic methodology was indeed new and not very reliable, and we have since made major efforts to improve it. We hope the reviewer is more convinced by the new datasets that show consistent behavioral effects of optogenetic inhibition in every single animal.

In addition, we used non-conditional chemogenetic silencing in DG where we observe a strong correlation between the expression of the Gi-DREADD and memory recall impairment after injection of clozapine-N-oxide (CNO). Of note, compared to the cFos-dependent optogenetic inhibition, chemogenetic inhibition of random subsets of granule cells required much larger numbers to be silenced before navigation was compromised. This shows for the first time the special importance of cFos-expressing granule cells for navigation.

2) The hippocampus is thought to provide cognitive maps of subjective experiences, and spatial coding is just one aspect of such maps. It is therefore not a priori clear that cFos ensembles likely reflecting memory should necessarily involve some explicit representation of space.

That is exactly the reason why we decided to test whether cFos ensembles in DG are more similar when the spatial behavior is more consistent. The answer is: No, there is no spatial information in the cFos pattern, at least not in the sense of a labeled line code (i.e. place cells).

3) The effects of the ensemble silencing experiments are very small, which renders the interpretation

of these unexpected findings difficult, particularly in view of current uncertainties as to the role of DG ensembles.

We agree and, as mentioned before, have improved our methods, resulting in clear and consistent effects of optogenetic inhibition. We confirmed these results in two different batches of mice; the silencing effect was significant in both batches (pooled analysis in Fig. 3h, i).

4) The authors use terms such as "active neurons" and cFos-expressing neurons as if they were interchangeable. There is no question that the expression of cFos depends on neuronal activity, but there is little reason to believe that ensembles of cFos expressing neurons exactly reflect ensembles of active neurons.

We absolutely agree and we regret the sloppy terminology we used in the original MS. We now use more precise and purely descriptive terms (cFos⁺) in the Results and debate the relationship between cFos and activity in the Discussion. Interestingly, it seems to be quite different in different subregions (DG vs CA1), and we present a candidate molecular mechanism underlying this difference (Δ fosB accumulation).

Reviewer #3 (Remarks to the Author):

The manuscript by Lamothe-Molina et al characterized how active ensembles in the dentate gyrus of mice change over time, even when mice are exposed to a constant environment or subjected to the same spatial navigation task over days. The authors used TetTag mice expressing a short half-life GFP under a cFos promoter to transiently tag neurons that were active within hours prior to histological analysis, along with viral injection of an activity-dependent inhibitory opsin (iChloc) to permanently tag and manipulate neurons that were active during a 24h window when mice were taken off a doxycycline diet. Permanent tagging with iChloc occurred in the home-cage or with water maze training.

Overall, the authors' data shows relatively little overlap between active neurons permanently tagged during spatial learning and neurons active during further training or testing 1d or more later.

Furthermore, optogenetic inhibition of neurons that were tagged during early learning did not impede performance when mice were later tested for spatial memory. These results suggest that cFos ensembles in the dentate gyrus change over time and do not represent a map of space.

While there is an abundance of intriguing data in this study and the conclusions are in line with another recent study by Tanaka et al (2018) in the CA1 region of hippocampus, I have a number of major and minor comments that should be addressed.

Major comments:

1. In Figure 1 the equivalent levels of GFP expression in HC (0d, 1d), as well as in the WM groups (0d, 1d) shows that GFP reflects background/baseline fluctuations in fos expression/neuronal activity (which is further confirmed in Figure 4). It is compelling to think that this population may reflect neurons within a given time frame from which ensemble/engram neurons can be chosen to encode the memory – supported by the fact that iChloc expression does not rise above GFP levels after WM training. However, as most studies in the DG (using fos, arc immunolabeling or other methods) show anywhere from 1-5% of cells become engram neurons, it means that anywhere from 50-90% of fos⁺ (or even iChloc⁺) cells are not part of the engram. This of course would have an impact on interpretation of the overlap calculations.

It would be critical, in my view, to be able to identify true engram neurons for overlap analysis, possibly by differences in fluorescence (as alluded to later in the manuscript). In addition, it is possible that the immunofluorescent amplification of signals masked any overt differences between low fos+ (non-engram) and high fos+ (engram) neurons.

We identified a problem in our reporter system: leakiness of the TREtight promoter resulted in faint expression even in on-doxycycline animals (Supplementary Fig. 3b). We realized that due to our highly sensitive method, we classified too many cells with very faint fluorescence as cFos+, resulting in a fraction of cFos+ granule cells (8-15% of DAPI) higher than reported in older studies. To correct for these issues, we have taken the following measures:

1. We investigated the origin of the unspecific tagging. We observed that even in mice that lack the tTA, there is some faint expression in the injected region. This suggests that the CMV region of the TREtight promoter from our iChloC construct was weakly activated by an unknown native transcription factor. We also observed some positive cells in mice TetTag mice (tTA+) that were always on Dox. We show these findings in the new Supplementary Fig. 3.
2. We reanalyzed every single image with the more stringent classification threshold. We re-blinded the analyst and shuffled the images to avoid any bias.
 - a. tTA::TRE-dependent cFos tagging (iChloC-mKate). To provide an objective criterium, we used On-Dox images to adjust the display settings for manual scoring. This way, weakly expressing neurons (promoter and doxycycline leakiness) were not scored as cFos+.
 - b. Short half-life cFos-reporter (shEGFP). Since this signal is localized to the nucleus, it was scored automatically using the Imaris spot function. For every staining batch, a classification threshold was set once and applied to all animals/slices from that batch.
3. We compared more precisely the shEGFP expression curve to the native cFos protein (new experiments in organotypic hippocampal slice culture at 37 °C). We confirmed that the shEGFP protein takes longer to rise. After 2 h, the correlation between reporter and native cFos is highly significant (Supplementary Fig 1), validating our strategy.
4. For behavioral experiments, we used a BiPOLES tool driven by the TRE3G promoter (no unspecific signal) and performed a new set of optogenetic manipulation experiments (see Supplementary Fig 3 and answers to reviewer #2).

The new high-threshold reanalysis, together with an increase in the number of mice of each group (replication experiments), resulted in a lower number of overlapping cells (Supplementary Fig. 2), but a greater contrast to chance levels (Fig. 1 and 2). Overlap/chance values increased in all WM-trained groups, now showing overlap significantly higher than chance for ET, OT, and RT groups (but not in HC and NE groups).

Analyzing the fluorescence intensity of shEGFP, we detected significant differences between WM-trained mice and home-caged mice (Fig. 2e). The resulting curves (cumulative histograms) illustrate that the differences between water maze groups and home cage are robust and independent of cFos scoring threshold. With the new analysis strategy, our cFos+ numbers (~4%) are now in line with previous studies (Reviewer Fig. 1, Supplementary Fig. 2).

Reviewer Figure 1: Old (low threshold) and new (high threshold analysis). a) New analysis results in low background detection (On Dox, upper right image). b) New analysis results in lower cFos counts for both home cage and water maze animals. c) Quantitative comparison of cFos ensemble size from low and high threshold analysis. Lines correspond to individual images randomly selected from all groups.

2. The optogenetic silencing experiment in Figure 2 is surprising and interesting. However, there is no certainty that iChloc is sufficiently tagging the relevant ensemble since overlap may not be more than 20% as suggested by Figure 4 (and some of these iChloc+ cells may be those tagged under basal conditions rather than WM training).

We agree and we use now a different promoter and more powerful opsin (see answers to reviewer #2). Now we find significant decrease in WM performance in trials with optogenetic inhibition of the day 1 cFos ensemble (Fig. 3).

Also, is 1 day of training (when mice haven't sufficiently learned) able to define the final memory ensemble on Day 3 or Day 5? I don't believe it is known if other neurons become fos+ as training proceeds on the second and third day etc. to add to the final memory ensemble – if so, that would suggest that optogenetic inhibition only inhibits a fraction of the memory ensemble on Day 5 and this is not due to time-dependent drift.

Indeed, the performance is still poor on day 1 of WM training, but with the new optogenetic tool, we do see a clear effect of inhibition of the day 1 ensemble even 5 days later. In the revised manuscript, we delineate how we interpret these results: In short, we believe that many neurons of the day 1 ensemble are the place cells tiling this specific environment. Later inhibition of this ensemble will not only block access to information about the platform position, but also interfere with the ability to represent the animal's own position in the WM environment. Both of these effects could contribute to the decrease in performance we observe.

Despite the significant effect of optogenetic inhibition, mice still showed preference for the target quadrant. We think that information stored by cFos⁺ cells on subsequent training days contributes to spatial memory.

3. It would also be a logical progression to perform the optogenetic inhibition on WM early training (ET) mice used in Figure 5, comparing d1 vs d2, as well as WM overtraining (OT) mice, comparing d5 and d6. As d5/d6 mice show surprisingly little overlap, possibly for reasons presented in the Discussion, it would also be useful to do a similar experiment comparing d2 and d3, d3 and d4, d4 and d5 (i.e. iChloc tag on d4, inhibit during training on d5).

In response to this comment, we tried silencing on day 2, 3, & 5 in the new BiPOLES experiments (tagging on day 1). Effects became significant on day 3 & 5. On day 2, inhibition had no significant effect, mainly because performance was still quite variable, making it difficult to detect small effects.

It would also be interesting to see if tagging on a day were mice are using more spatial strategies has a bigger impact on performance. The original aim of this study was to emulate as close as possible what other experiments using contextual fear conditioning use. Therefore, we focused on the cFos ensemble from day 1, the first day of water maze training, equivalent to the first exposure to the context in CFC experiments.

4. The above comment additionally highlights a need for a positive control for optogenetic inhibition of behavior. Contextual fear conditioning would be ideal as this has been previously shown.

This comment refers to the former experiments using iChloC. In the new experiments (BiPOLES), we see a clear effect of optogenetic inhibition, making further positive controls unnecessary. Negative controls are the trials without optogenetic inhibition that we recorded in each mouse.

5. Figure 4e is the most convincing evidence for shGFP turnover within hours, but there are not enough data points. I suggest adding 12h and 18h groups. It feels like wishful thinking to describe this as a sigmoidal function based on 3 data points.

We performed additional experiments and added a $\Delta t = 12$ h group. Instead of fitting mathematical functions, we now use a modeling approach. A numerical simulation simulating delayed Δ FosB accumulation and block of cFos expression reproduced very nicely the linear drop in overlap/chance we observed experimentally in DG, including the drop *below* chance level at $\Delta t = 24$ h. Interestingly, the same model worked for CA1 if we removed the negative feedback loop.

6. The *in vitro* data showing shGFP turnover (Extended Data Figure 1d) is not completely clear to me. As far as I understand it, whole-image fluorescence peaks 3h after stimulation and returns to almost baseline by 6h. Does that leave open the possibility that there is still low fluorescence that can be detected? This figure also shows that fluorescence in “spots” peaks at 3h and stays level. What are spots? The Methods section seem to imply the diameter of these spots correspond to cells – if that is the case, then number of fluorescent cells doesn’t go down. How does this fit with the whole image fluorescence? I also note that one graph goes to 5h while the other goes to 6h on the x-axis.

The original experiments were done at room temperature, which was not ideal. We repeated the experiments at 37 °C and using immunofluorescence staining instead of live imaging. We fixed organotypic cultures at different timepoints after high KCl stimulation and correlated anti-EGFP to anti-cFos fluorescence in individual GCs. The time course of the new experiments shows rise and decay of cFos and its reporter under physiological conditions (Supplementary Fig. 1).

7. In a number of experiments, the number of mice used is low (e.g. Figure 1e: 3 mice in some groups, 7 mice in others; Figure 6d: n=4 and 5). This reduces the statistical power of these experiments.

We have now at least 6 mice for all the cFos overlap analysis in the WM groups. For the temporal shift groups in HC, 3 mice per group were sufficient since we found strongly significant differences (large effect size). For the new optogenetic silencing experiments, we tested 15 mice (paired statistics). All animals were tested with and without light in two consecutive probe trials. We did these experiments in two different batches with similar methodology, the effect was significant in both batches.

8. According to the Methods, the mice used were 20-60 weeks of age (3.5-14 months, approximately). This is a huge variation and there have been studies that show alterations in hippocampal synaptic plasticity and memory in aged mice (>1yr). As such, it would be incumbent for the authors to examine the behavior and fos/iChloc overlap in 20-week-old vs 60-week-old mice and show no difference, or account for any differences when performing their analysis.

Most of the older mice were from the animals used for the iChloC experiments. For the new BiPOLES experiments, we used mice not older than 7 months, reducing the variability.

9. Statistical analyses. Important details of statistical analyses are missing. For example, the data in Fig. 3e were analyzed by mixed 2-way ANOVA, but main effects and interactions are not reported. Post-hocs (such as the Tukey post hoc test reported here) should only be reported following a significant interaction term. This same criticism applies the data in 3f, 5c.

We have now added every test to the figure legends. Following ARRIVE guidelines, we provide a supplementary table with all details.

10. The argument that these data provide evidence that the DG is doing some form of temporal pattern separation or time stamping is interesting. However, this might also be true of any region in the brain. For example, would the authors expect the same results were they to look at other regions processing spatial memory information such as CA1, retrosplenial cortex and so on. The current arguments would be greatly strengthened were the authors able to show that this is only happening in the DG.

To address this criticism, we now compare DG to other hippocampal areas. We show that the Δ FosB to cFos blocking mechanism is specific to the GCs of DG (Fig. 5a-g). Imaging hippocampal slices from expert WM mice, DG showed very high density FosB immunoreactivity in comparison to CA3, CA1 and subiculum (Fig. 5i). At least in cortical regions close to dorsal hippocampus, DG seems to be unique in this respect. Furthermore, we now show that the cFos temporal dynamics in vivo are very different in DG and CA1 (Fig. 4). Altogether, these differences fit well to the proposed function of DG, pattern separation and formation of episodic, time stamped memories.

Minor comments:

9. The schematic of shGFP expression in all the figures is somewhat misleading – it suggests a specific population shows gradually increased GFP expression which then reduces to 0 over 6h, and then another population takes over (and thus there is a period with no or low GFP expression), rather than neurons that become fos+ and other neurons that become fos- over constant time periods. I realize the schematic is to illustrate a larger idea and so perhaps the reality should at least be acknowledged.

We agree, it has now been removed from all figures.

10. In Figure 1b, it appears that different high magnification images were used for mKate2, shGFP and overlap – it would be better to use one image in the different channels to show the sort of expression one expects.

Images were changed now. All comparative images are now the same size and shown in individual channels.

11. What is “adequate expression” for the inclusion/exclusion criteria of virus infection (Methods, p33)? This should be detailed further.

We have corrected that statement in the methods. We exclude mice that show expression in only one of the blades of the DG.

12. I’m not sure prototypical fear conditioning is primarily considered a model for PTSD as suggested on p23 of the Discussion.

This is no longer in the text.

13. There are some typos, formatting errors and at least one mismatch between papers cited in the main text and the References section.

We have corrected this.

Reviewers' comments:

Reviewer #1 (Remarks to the Author):

The authors have done a tremendous job addressing my concerns -- this is an extensively revised manuscript both in terms of new experiments, an updated scholarly discussion, and a effortful contextualization of their findings within the context of the neuroscience literature. The manuscript in my opinion is suitable for publication at Nat Comms and will be a significant advancement for the field as a whole. It is a much-needed bridge between engram-tagging strategies (e.g. activity-dependent manipulations) and spatial memory processes with novel findings on the mechanistic basis of the latter.

Reviewer #2 (Remarks to the Author):

The authors have addressed the points raised in my previous review and have clarified several of the previous issues. As a result, the main story in the paper has changed substantially. This largely represents an improvement but the new focus comes with new issues that need to be addressed. In addition, the current text does not always do justice to the new findings, particularly in abstract and title.

Specific points:

1) One major point is whether day1 now represents a singular day with respect to water maze learning and establishment of ensembles in the dentate, or whether each day makes an essential contribution to the learning process as reflected in the dentate granule cell ensembles. The authors show that silencing the day1 ensemble on day5 interferes with reference memory. Given the fact that cFos+ ensembles change dramatically every day in the dentate the authors should provide additional data addressing the impact of silencing the day2 ensemble on day5 (day1 unique versus each day essential contribution?) and silencing the day4 ensemble on day5 (still large impact or now smaller impact because most of the task learned?).

2) A key new finding involves the appearance of FosB/ Δ FosB in granule cells. The authors highlight the possible role of Δ Fosb, which would be exciting, but the argument is largely speculative so far, and, the current data need to be strengthened. The authors need to provide experiments in which they stimulate chemogenetically DG or CA1 neurons (slices) and detect FosB/ Δ FosB on the next day in the absence of stimulation. Is the late appearance in CA1 due to late expression, or due to stimulation on day2? A second key piece of missing data would be any evidence that the granule cells indeed accumulate inhibitory Δ FosB 24h after stimulation (or learning), as inferred, as opposed to FosB - could this be addressed by immunoblots or any specific reagent?

3) The FosB/ Δ FosB data in situ (Fig. 5e) are sub-par. The authors should also provide panels with single labelling (as opposed to only merged images) in order to more convincingly show that the granule cells that expressed cFos express FosB/ Δ FosB, whereas the CA1 neurons do not. In addition, how about the massive expression on day2 in CA1 (slice data): are these neurons that expressed cFos on day1?

4) The title might have to be modified to more explicitly express the main findings. The relation to a spatial map is speculative. In addition, the abstract should at the very least highlight the distinction between DG and CA1 with respect to re-expression of cFos and expression of FosB/ Δ FosB. If not incompatible with length restrictions, the abstract should be extended so that casual readers pick up the main findings with their potential explanations, as opposed to a series of puzzling statements.

Reviewer #3 (Remarks to the Author):

The manuscript by Lamothe-Molina et al characterized how fos-expressing ensembles in the dentate gyrus of mice change over time, even when mice subjected to the same spatial navigation task over days. The authors used TetTag mice to enable both transient and longer-term tagging of fos+ neurons, the latter limited by a temporal window that opens by taking mice off a doxycycline diet. Analysis of tagged fos+ neurons during different days of training in a water-maze task suggests that cFos ensembles in the dentate gyrus generally change from day to day of water-maze training and thus do not represent a stable map of space. Interestingly, for neurons tagged on day 1 of training, the lack of fos activation in subsequent days appears to be related to negative feedback caused by accumulation of FosB/Fos Δ B and, despite the lack of subsequent fos activation, optogenetic inhibition of these neurons in later days of training still reduced performance in the task.

The authors have adequately addressed the bulk of my concerns in the initial review and the resulting manuscript is compelling and convincing. It would be extremely interesting if the authors could add an experiment where they tagged fos+ neurons on later days of training and tried the optogenetic inhibition experiment to see if the sub-assemblies activated on later days in the task are also necessary for memory retrieval. However, I believe the manuscript still stands well as it is.

Reviewer #1 (Remarks to the Author):

The authors have done a tremendous job addressing my concerns -- this is an extensively revised manuscript both in terms of new experiments, an updated scholarly discussion, and a effortful contextualization of their findings within the context of the neuroscience literature. The manuscript in my opinion is suitable for publication at Nat Comms and will be a significant advancement for the field as a whole. It is a much-needed bridge between engram-tagging strategies (e.g. activity-dependent manipulations) and spatial memory processes with novel findings on the mechanistic basis of the latter.

We are very thankful for the positive comments throughout the review process!

Reviewer #2 (Remarks to the Author):

The authors have addressed the points raised in my previous review and have clarified several of the previous issues. As a result, the main story in the paper has changed substantially. This largely represents an improvement but the new focus comes with new issues that need to be addressed. In addition, the current text does not always do justice to the new findings, particularly in abstract and title.

We changed abstract and title to better reflect the new focus on cFos regulation.

Specific points:

1) One major point is whether day1 now represents a singular day with respect to water maze learning and establishment of ensembles in the dentate, or whether each day makes an essential contribution to the learning process as reflected in the dentate granule cell ensembles. The authors show that silencing the day1 ensemble on day5 interferes with reference memory. Given the fact that cFos+ ensembles change dramatically every day in the dentate the authors should provide additional data addressing the impact of silencing the day2 ensemble on day5 (day1 unique versus each day essential contribution?) and silencing the day4 ensemble on day5 (still large impact or now smaller impact because most of the task learned?).

This question about the unique function of the day 1 ensemble is very interesting. We include two new figures in the revised MS that we hope shine some light on this question: In Fig. 4 we use the calcium integrator CaMPARI2 *in vivo* to show that day 1 cFos neurons are indeed active (high calcium levels) on the next day of WM training, but not when exposed to a different environment. This confirms our suspicion that the day 1 cFos cells are place- or task-specific. Fig. 8 shows a new analysis comparing spaced training (day 1 and day 7) to daily training. Indeed, with more training, we find more and more granule cells in the inhibited state (high Δ FosB), indicating that there is accumulation of Δ FosB on every training day. We conclude that day 1 is unique as there is little or no inhibition of cFos expression by transcriptional mechanisms (blank slate), and that every training day further increases the number of granule cells in the repressed state (see Fig. 8 and Supp. Figs. 10 and 11).

2) A key new finding involves the appearance of FosB/ Δ FosB in granule cells. The authors highlight the possible role of Δ Fosb, which would be exciting, but the argument is largely speculative so far, and, the current data need to be strengthened. The authors need to provide experiments in which they stimulate chemogenetically DG or CA1 neurons (slices) and detect FosB/ Δ FosB on the next day

in the absence of stimulation. Is the late appearance in CA1 due to late expression, or due to stimulation on day2?

We completely agree. We have included a new dataset (Supp. Fig. 9) where we stimulated (Gq) slice cultures with CNO and looked for panFosB immunoreactivity at 24 and 48 h after stimulation. We show that indeed panFosB levels are elevated 24 h later but decrease after 48 h. As expected, cFos levels were low at 24 and 48 h. These data suggest that the observed high panFosB IR after the 2nd CNO application (Fig. 7) is coming from day 1 CNO stimulation and not from the 2nd stimulation.

A second key piece of missing data would be any evidence that the granule cells indeed accumulate inhibitory Δ FosB 24h after stimulation (or learning), as inferred, as opposed to FosB - could this be addressed by immunoblots or any specific reagent?

In the aforementioned experiments, we used an AB that recognizes all variants of the FosB protein (pan-FosB, N-terminal region epitope), so we could not tell whether Δ FosB was responsible for the block of cFos expression. To address this weakness, we have used a new approach combining the pan-FosB AB with a second AB that recognizes the C-terminal region of FosB (which is missing in Δ FosB). We used this method to analyze Δ FosB IR in WM-trained mice (new Fig. 8, Supp. Figs. 10-12). We decided to go for an approach that could detect the protein levels rather than mRNA since we hypothesized that the accumulation of Δ FosB protein is responsible for cFos expression repression. Immunoblots would not have given us the required cell-type specificity.

3) The FosB/ Δ FosB data in situ (Fig. 5e) are sub-par. The authors should also provide panels with single labelling (as opposed to only merged images) in order to more convincingly show that the granule cells that expressed cFos express FosB/ Δ FosB, whereas the CA1 neurons do not. In addition, how about the massive expression on day2 in CA1 (slice data): are these neurons that expressed cFos on day1?

Thank you for the suggestion, we now show the fluorescence channels in separate panels (Fig. 6a, c). For clarity, we added a schematic drawing to show the overlap between cFos day1 with cFos day7/pan-FosB for the example (Fig. 6c). We have added new images to show cFos and pan-FosB IR in the whole DG region together with a correlation graph of the fluorescence intensity of both signals without considering the cFos-tagged cells from day 1 (Fig. 6b, c). We observed a clear inverse correlation, suggesting that repeated cFos expression is blocked by a member of the FosB protein family. In order to test if the responsible protein is Δ FosB, we used the same double AB approach as in the slice experiments (see above) and found that the inverse correlation is indeed by Δ FosB (Fig. 8, Supp. Fig. 10, 11). We additionally performed this analysis in mice that were not trained at all (home cage) and in mice that received spaced WM training. We found that Δ FosB accumulates in GCs when mice are trained in the WM and that spaced training led to fewer cFos-repressed GCs than daily training.

4) The title might have to be modified to more explicitly express the main findings. The relation to a spatial map is speculative. In addition, the abstract should at the very least highlight the distinction between DG and CA1 with respect to re-expression of cFos and expression of FosB/ Δ FosB. If not incompatible with length restrictions, the abstract should be extended so that casual readers pick up the main findings with their potential explanations, as opposed to a series of puzzling statements.

We completely agree, experiments from Figs. 5-7 extended the scope and shine some light on cFos regulation in GCs, also demonstrating the absence of this regulation in CA1. We have changed the

title of the manuscript to “ Δ FosB accumulation in hippocampal granule cells drives cFos pattern separation during spatial learning” and updated introduction and discussion.

Reviewer #3 (Remarks to the Author):

The manuscript by Lamothe-Molina et al characterized how fos-expressing ensembles in the dentate gyrus of mice change over time, even when mice subjected to the same spatial navigation task over days. The authors used TetTag mice to enable both transient and longer-term tagging of fos+ neurons, the latter limited by a temporal window that opens by taking mice off a doxycycline diet. Analysis of tagged fos+ neurons during different days of training in a water-maze task suggests that cFos ensembles in the dentate gyrus generally change from day to day of water-maze training and thus do not represent a stable map of space. Interestingly, for neurons tagged on day 1 of training, the lack of fos activation in subsequent days appears to be related to negative feedback caused by accumulation of FosB/Fos Δ B and, despite the lack of subsequent fos activation, optogenetic inhibition of these neurons in later days of training still reduced performance in the task. The authors have adequately addressed the bulk of my concerns in the initial review and the resulting manuscript is compelling and convincing.

We are pleased to hear all the concerns were addressed and we want to thank the reviewer for the positive comments.

It would be extremely interesting if the authors could add an experiment where they tagged fos+ neurons on later days of training and tried the optogenetic inhibition experiment to see if the sub-assemblies activated on later days in the task are also necessary for memory retrieval. However, I believe the manuscript still stands well as it is.

Please see our answer to the first point of reviewer #2.